# Vision Language Models Cannot Reason About Physical Transformation

**Dezhi Luo** [* 1]  **Yijiang Li** [* 2]  **Maijunxian Wang** [3]  **Tianwei Zhao** [4]  **Bingyang Wang** [5]  **Siheng Wang** [6]
**Pinyuan Feng** [7]  **Pooyan Rahmanzadehgervi** [8]  **Ziqiao Ma** [1]  **Hokin Deng** [9]

## Abstract

Understanding physical transformations is fundamental for reasoning in dynamic environments. While Vision Language Models (VLMs) show promise in embodied applications, whether they genuinely understand physical transformations remains unclear. We introduce *Conservation-Bench* evaluating *conservation*—whether physical quantities remain invariant under transformations. Spanning four properties with paired conserving/non-conserving scenarios, we generate and evaluate 23,040 questions across 112 VLMs. Results reveal systematic failure: performance remains near chance with improvements on conservation tasks accompanied by drops on controls. Control experiments show strong textual priors favoring invariance, yet models perform worse with actual visual content when performance is balanced across conserving and non-conserving scenarios. Neither temporal resolution, prompting, nor curated sampling helps. These findings show that current VLMs fail to maintain transformation-invariant representations of physical properties across dynamic scenes.

## 1. Introduction

Recent advances in Vision Language Models (VLMs) (Zhang et al., 2024c; Radford et al., 2021; Alayrac et al., 2022; Li et al., 2023) have demonstrated remarkable capabilities of perception (Wang et al., 2024; Chen et al., 2025; Jiang et al., 2025; Team et al., 2025; Cheng et al., 2024b), reasoning (Zhang et al., 2024b; Xu et al., 2024; Cheng et al., 2024a), and visual commonsense

understanding (Zellers et al., 2019; Park et al., 2020). These capabilities hold promise for real-world applications (Brohan et al., 2023), particularly in embodied tasks (Driess et al., 2023; Nasiriany et al., 2024) that demand a genuine understanding of the physical world and its underlying properties (Chow et al., 2025b; Gao et al., 2024a). Yet it remains unclear whether VLMs possess a true understanding of physical principles or the capacity to operate reliably in embodied physical environments.

A key factor in human intelligence that enables successful navigation in an embodied, physically grounded world is the ability to understand and reason about physical transformations (Piaget, 1950; 1952; 1965; Baillargeon et al., 1985; 1990; Baillargeon, 1987; 1986; Spelke et al., 1992; Baillargeon & Carey, 2012; Bear et al., 2021; Piloto et al., 2022). This capacity includes tracking objects over time (Spelke et al., 1994; 1995), managing occlusions (Gredebäck & von Hofsten, 2004), and adapting to dynamic environments (Allen et al., 2020). While there are benchmarks evaluating physically plausible video generation (Motamed et al., 2025; Meng et al., 2024; Yang et al., 2025; Liu et al., 2025; Shi et al., 2024) and physical understanding in VLMs, spanning from everyday scenes (Zheng et al., 2024; Chow et al., 2025a) to high-school physics questions (Wang et al., 2025) and Olympiad-level problems (Qiu et al., 2025; Wang et al., 2025), these efforts focus either on video generation or physical properties in static scenes, leaving underexplored whether VLMs can genuinely reason about physical transformations—where specific properties may or may not remain invariant.

To bridge this gap, we evaluate *conservation* in VLMs—the understanding that physical quantities remain invariant under transformation despite changes in appearance. Here, physical quantity refers to the measurable magnitude of objects along certain dimensions, while spatial transformation denotes the continuous process through which objects change in appearance or position. For example, an agent demonstrating conservation would recognize that pouring water into a differently shaped glass does not alter its volume, despite the change in visible form. Achieving conservation thus requires more than linguistic knowledge of quantity: it demands a systematic understanding that is both reversible and grounded in visual as well as conceptual

---
[*]Equal contribution  [1]University of Michigan  [2]University of California San Diego  [3]University of California Berkeley  [4]Johns Hopkins University  [5]Georgia Institute of Technology  [6]University of Toronto  [7]Columbia University  [8]Auburn University  [9]Carnegie Mellon University. Correspondence to: Dezhi Luo <ihze-doul@umich.edu>, Yijiang Li <yijiangli@ucsd.edu>.

*Proceedings of the 43ʳᵈ International Conference on Machine Learning*, Seoul, South Korea. PMLR 306, 2026. Copyright 2026 by the author(s).

| Task | Question | Conserving | | | Non-conserving | | |
|------|----------|:----------:|---|---|:--------------:|---|---|
| **Number** | Is the number of coins in the upper row the same as in the lower row in the final image? | | | | | | |
| **Length** | Is the length of the upper straw the same as the length of the lower in the final image? | | | | | | |
| **Size** | Is the size of the playdough in the first image the same as in the final image? | | | | | | |
| **Volume** | Is the amount of liquid in the left glass in the first image the same as in the right in the final image? | | | | | | |

*Figure 1.* Illustrative Tasks and Frame Selection Pipeline in *Conservation Bench*.

representations. We introduce *ConservationBench*, a cognitively grounded benchmark for evaluating whether VLMs can reason about physical transformations. The benchmark consists of 192 video-based tasks across four core quantitative properties (number, length, volume, and size), each requiring models to judge whether a quantity is conserved despite visual transformations. To control for shortcut exploitation, we include 192 matched non-conserving controls where the target quantity changes while irrelevant features remain constant. We systematically vary frame extraction method, temporal resolution, and prompting strategy, yielding 60 conditions and 23,040 total trials.

Evaluating 112 VLMs, we find that models consistently fail to integrate temporal information to track conserved properties across dynamic scenes. High accuracy on conservation tasks is often driven by default heuristics, which reverse in non-conserving scenarios, revealing brittle, non-generalizable reasoning. Furthermore, prompting with cues encouraging transformation reasoning or providing higher temporal resolution does not help. These findings expose a fundamental limitation in current VLMs and underscore the need for more grounded, temporally-aware models capable of systematic physical inference.

## 2. Related Works

**Evaluating VLMs.** Early benchmarking efforts relied on single-task benchmarks such as VQA (Antol et al., 2015), OK-VQA (Marino et al., 2019), and OCR (Liu et al., 2023). However, with the emergence of VLMs that claim broader perceptual and reasoning abilities, evaluation has shifted toward holistic benchmarks such as MMMU (Yue et al., 2024), SEED-Bench (Li et al., 2024a), and MMBench (Liu et al., 2024). A growing line of benchmarks focuses specifically on quantity understanding (Rane et al., 2024; Paiss et al., 2023; Rahmanzadehgervi et al., 2024; Yuksekgonul et al., 2022). These tasks typically assess a model's ability

to individuate and count discrete objects in static scenes. While useful, such evaluations largely reduce to surface-level enumeration and do not test whether models encode numerical invariance—invariance of quantity across transformations. In contrast, our work examines whether VLMs go beyond perceptual counting to represent quantity as a transformation-invariant property.

**Physical Understanding and Conservation.** Insights from cognitive science underscore conservation as a critical benchmark for systematic physical reasoning. First proposed by Piaget, success on conservation tasks has long been viewed as evidence of emerging mental operations (Piaget & Inhelder, 1969). Developmental studies show that solving these tasks requires constructing transformation-invariant representations while suppressing misleading perceptual cues (Goldin-Meadow & Beilock, 2010; Houdé et al., 2011; Poirel et al., 2012). Behavioral and neurocognitive research further demonstrates that conservation performance depends on sensorimotor grounding and inhibitory control, highlighting the embodied nature of transformation understanding (Beilock & Goldin-Meadow, 2010; Lozada & Carro, 2016). Conservation also builds on more rudimentary abilities such as object permanence and individuation, revealed through studies exploiting the tunnel effect and violation-of-expectation paradigms (Burke, 1952; Flombaum & Scholl, 2006; Noles et al., 2005; Scholl, 2007), which themselves provide essential foundations for robust physical reasoning. In this light, conservation is widely recognized as a fundamental cognitive capacity for the higher-level physical reasoning needed to navigate dynamic, embodied environments (Fodor, 1975; Baillargeon & Carey, 2012; Barsalou, 2020; Luo et al., 2025b).

Recent studies have examined models' abilities to reason about physical properties, causal interactions, and material dynamics (Chow et al., 2025a; Patel et al., 2022; Zheng et al., 2024; Li et al., 2025a). Growing evidence suggests that VLMs struggle with fundamental aspects of visual rea-

soning and physical understanding (Campbell et al., 2024; Gao et al., 2024b; Sun et al., 2024; 2025; Gao et al., 2025; Schulze Buschoff et al., 2025; Buschoff et al., 2025), with some work exploring how modular frameworks or synthetic training data might address these limitations (Balazadeh et al., 2024; 2025; Luo et al., 2025b). However, these efforts largely emphasize outcome prediction or descriptive inference, without testing whether models recognize that certain properties remain invariant under transformation. In many cases, success appears to stem from outcome-based heuristics rather than structured mental operations (Newman et al., 2024; Isola et al., 2015). Consequently, it remains unclear whether current VLMs can genuinely integrate sequential evidence to track physical transformations while maintaining stable representations of underlying properties—a core cognitive capacity directly targeted by conservation tasks (Mitchell & Krakauer, 2023).

## 3. Experimental Design

### 3.1. Conservation Tasks

To systematically measure the conservation ability of VLMs– the understanding that specific physical properties remain invariant under transformations despite changes in appearance, we construct a suite of conservation tasks in the form of videos that visually depict physical transformations across four fundamental quantitative properties. We illustrate conservation tasks across each property in Figure 2, with full descriptions provided in Appendix B. Although the four conservation types probe distinct physical properties, the tasks follow a unified structure: a transition from an initial to a final state mediated by an observable transformation. Each video begins with an initial state, proceeds through a continuous transformation (e.g., pouring, spreading, flattening), and ends with a new state where the surface appearance of the object of interest is altered. This design mirrors real-world scenarios where physical reasoning depends on integrating perceptual evidence across time.

**Generalization across Task-irrelevant Features.** To ensure the robustness and generalizability of the conclusions drawn from our benchmark, we systematically vary key visual parameters in each conservation task (Table 4). These parameters include object count, size, color, layout, container shape, and the direction of transformation. Each conservation property consists of 48 unique video instances of different configurations, resulting in a total of 192 videos. This controlled variation guarantees that the core conservation principle is preserved across a wide range of visual contexts, thus preventing models from relying on memorized templates or superficial cues.

**Transformation-mandatory vs. -helpful.** Notably, conservation tasks differ in how strongly they depend on observing the transformation. We classify them into two categories:

*transformation-mandatory* and *transformation-helpful*. In mandatory tasks (volume and size), witnessing the transformation is essential—for instance, in volume conservation, seeing the liquid poured is necessary, since the final height alone is insufficient for judging quantity. In helpful tasks (number and length), correct judgments can still be made from the initial and final states, as the relevant quantity remains visually accessible despite superficial changes. This distinction enables a more diagnostic evaluation: models that excel on helpful but not on mandatory tasks may rely on static cues rather than forming internal representations of the process.

### 3.2. Non-conserving Tasks

A key limitation of applying conservation tasks to model evaluation is the uniformity of ground-truth labels: since all standard tasks involve quantity preservation, models can appear accurate simply by defaulting their responses to indicate invariance, due to biases from either visual contexts or linguistic patterns in the prompts, without genuinely reasoning about the physical transformation itself (Li et al., 2025b). To address this, we create non-conserving counterfactuals as a set of controlled experiments where the quantity of interest is explicitly altered during the transformation without changing the task-irrelevant features. That is to say, these manipulations are performed within the same environments, using identical object sets and visual contexts, thereby ensuring a controlled comparison. This design enables fine-grained assessment of model sensitivity to actual changes in quantity, rather than reliance on superficial heuristics or distributional priors. Details regarding control tasks across each property are available in Figure 2, with full descriptions provided in Appendix B. Following this design, we curated a control set in which each non-conserving control is paired with a conservation task under matched configurations, yielding an additional 192 videos.

### 3.3. Adaptation to Multi-frame Input

**Temporal Resolution** The ability to understand physical transformations critically depends on comprehending dynamic processes over time. Unlike static snapshot reasoning, robust comprehension requires recognizing continuity across successive observations. Human perception benefits from high frame rates (e.g. $\sim$ 30-60 frames per second) that convey rich temporal information, while the architectural and computational limitations of VLMs restrict them to inferring such dynamics from discrete and often sparse inputs. To investigate the impact of temporal resolution on conservation understanding, we vary the number of frames extracted from each video:

- **3-frame condition**: Only three frames are provided—the first, the last, and one intermediate frame. This condition presents minimal temporal information

while retaining just enough cues for humans to solve the task.

- **5-, 7-, and 9-frame condition**: More frames are sampled to offer moderate temporal granularity. This condition is designed to contrast qualitatively with the 3-frame condition by enabling multi-frame representations of the temporally continuous scene.

- **16-frame condition**: Sixteen frames are sampled to provide finer-grained temporal information, offering a more detailed depiction of the transformation process, contrasting quantitatively with the 8-frame condition.

All conditions include initial and final states. This design tests whether models can leverage higher temporal resolution to extract transformation-relevant information for conservation reasoning.

**Sampling Strategy** In studying physical transformations, the sequence and selection of visual inputs are crucial. This raises an important question: do different frame selection strategies influence the model's understanding of dynamic scenes? Additionally, do humans and models rely on different criteria when identifying informative visual moments? To examine this, we implement and compare three frame extraction strategies, each reflecting distinct assumptions about what defines a "representative" moment in a physical event.

- **Uniform Sampling**: Frames are sampled uniformly across the timeline, serving as a baseline approach commonly used in prior work, based on the assumption that temporal regularity sufficiently represents informational diversity.

- **Human-based**: To obtain a baseline for human intuition in frame extraction, we recruited N = 18 annotators. Each annotator was randomly assigned a subset of the dataset and asked to manually select the intermediate frames that captured the essential stages of the transformation.

- **Model-based**: We adopt SEVILA (Yu et al., 2023) and leverage a BLIP-2-based Localizer to identify language-aware keyframes. Prompted with the same instruction assigned to humans ("extract the most complete set of frames that capture the entire process"), the Localizer module selects frames with high relevance scores, which are then passed to the Answerer module for inference. This method formalizes a strategy akin to semantic salience: choosing frames that are maximally informative given a specific query.

This design allows us to test whether different frame selection strategies affect model performance on physical transformation reasoning. We hypothesize that optimizing frame selection, rather than merely increasing frame quantity, leads to more effective representations of dynamic events. We detail our data curation process in Appendix A and prompting strategies in Appendix C, and provide example input in Appendix D.

*Table 1.* Overview of Multi-image Task Conditions and Evaluation Scale

| Component | Count |
|---|---|
| *Core Dataset* | |
| Conservation Tasks | 192 |
| Non-conserving Control | 192 |
| Total Videos | 384 |
| *Multi-frame Conditions (Factorial)* | |
| Extraction Method | 3 |
| Frame Count | 5 |
| Prompting | 4 |
| Factorial Combinations | $3 \times 5 \times 4 = 60$ |
| **Total Evaluation Trials** | $384 \times 60 = 23,040$ |

## 4. Experiments

### 4.1. Inference and Evaluation

**Inference.** We evaluate 112 VLMs spanning diverse model architectures, training data, and parameter scales, covering both mainstream commercial systems and advanced open-source models. To ensure fidelity, comparability, and reproducibility, we strictly adhere to reference configurations and implementations from the official codebases. Refer to Appendix E for further details.

**Evaluation.** Each task presents three answer choices: for example, in number tasks, options are "No, the lower row has more coins," "No, the upper row has more coins," and "Yes, they are the same." Answer option order was shuffled and counterbalanced across all trials to eliminate positional bias. To evaluate free-form outputs of VLMs on multiple-choice questions (MCQs), we follow the two-stage scoring method of Li et al. (2025a). In Stage 1, each VLM output is mapped to a unique choice from the provided options or labeled FAIL when no unambiguous mapping is possible. Mapping follows a hybrid strategy: deterministic template matching is applied first, and unresolved cases are adjudicated by an LLM-as-a-Judge constrained to the option set. Models exhibiting persistently high FAIL rates are excluded from further analyses to avoid bias from nonsensical outputs. In Stage 2, the mapped option is compared against the ground-truth answer, with all FAILs scored as incorrect. Details are provided in Appendix F.

### 4.2. Human Baseline

Given the large number of questions and the cost of human annotation, we curated a representative subset by ran-

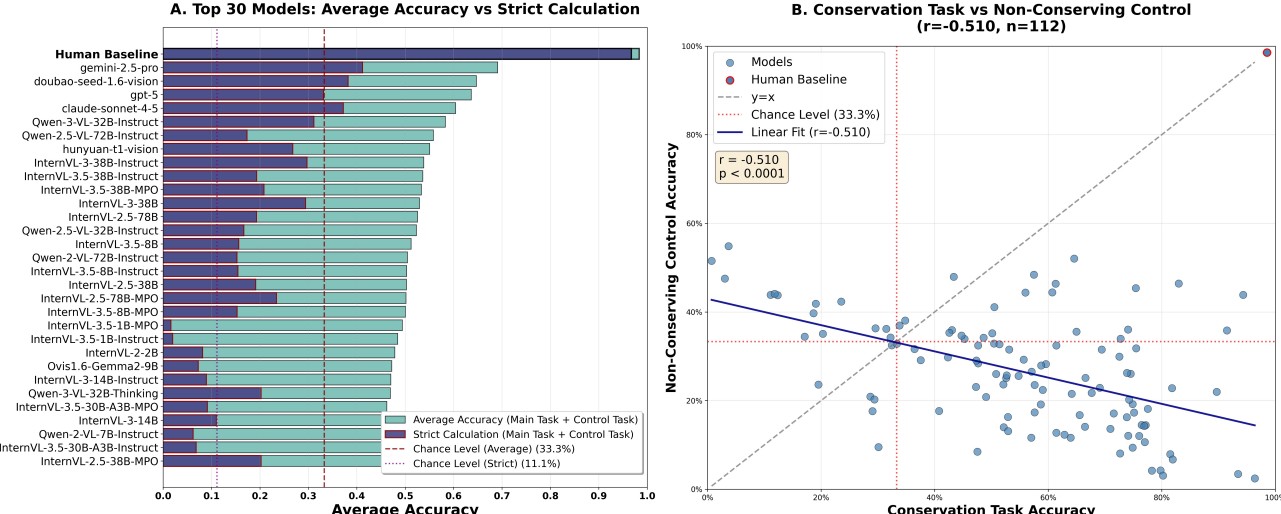

*Figure 2.* Overall Performance on *ConservationBench*. **A.** Accuracy averaged across conservation tasks and non-conserving controls, compared to strict pairwise accuracy (purple bars), which requires a model to answer *both* a conservation task and its matched non-conserving control correctly to be counted as correct (Top 30 models; full results in Appendix H). **B.** Performance on non-conserving controls in relation to conservation task accuracy across all 112 VLMs.

domly selecting one out of every eight task configurations for each quantitative property, counterbalanced across conservation tasks and non-conserving controls, resulting in 864 questions total. We recruited 7 participants online through a professional data-collection platform with standardized quality-control procedures. One participant was excluded for chance-level performance on a consecutive block of questions, indicating inattention, leaving 6 participants for analysis. Participants received the same visual and textual stimuli as the VLMs through a human-friendly interface and directly selected answers without any LLM-based answer parsing. The aggregated human accuracy reaches 98.35%, consistent with decades of developmental research showing that humans from late childhood reliably solve conservation tasks with near-perfect accuracy (Piaget, 1965; Houdé, 1997; Pezzulo et al., 2013; Viarouge et al., 2019). These results validate our benchmark design and its adaptation for evaluating VLMs. Detailed breakdown and comparison with models are highlighted in Appendix H.

### 4.3. Main Results

As shown in Figure 2A, model accuracy across 112 VLMs ranges from ∼20% to ∼69%, with most performing only marginally above the 33.3% chance level. In contrast, human participants exceed 98% accuracy, highlighting a clear gap between VLMs and intuitive human reasoning. Collectively, these results reveal a core limitation: VLMs struggle to integrate temporal cues or track invariant properties through dynamic transformations, a key requirement for grounded physical reasoning. We report the performance of all models and human baseline in Appendix H.

**Non-Conserving Control Reveal Systematic Bias**  By comparing model performance on non-conserving control tasks against conservation tasks, we observe a moderate negative correlation ($r = -0.510$, $n = 112$): models that perform better on conservation tasks tend to perform worse on the corresponding control tasks, and vice versa (Figure 2B). Most models cluster in the lower-right quadrant, exhibiting moderately high conservation accuracy (40–80%) but low non-conserving accuracy (10–40%), revealing a systematic bias toward quantity invariance regardless of actual transformation evidence. Only a small subset of models approaches balanced performance near the diagonal ($y = x$), while virtually none achieve high accuracy on both task types simultaneously. This pattern generalizes across four quantitative domains (see Appendix I for details). Crucially, this pattern reveals a diagnostic failure: models are not simply underperforming but exhibiting asymmetric reliance on default heuristics that systematically reverse across matched task conditions, demonstrating an inability to flexibly adjust reasoning based on transformation evidence.

We further validate this pattern using a *strict pairwise evaluation* across the full set matched conservation and non-conserving control tasks (Figure 2A; labeled in purple). In this analysis, a model is marked correct only if it answers both tasks in a pair correctly—capturing whether it can jointly recognize quantity preservation and detect meaningful violations under matched visual conditions. We find that most models (82/112, 73.2%) perform well below chance, achieving strict accuracy rates under 10%. Only three top-performing models—GEMINI-2.5-PRO, DOUBAO-SEED-1.6-VISION, and CLAUDE-SONNET-4-5—exceed chance level (33.3%). This indicates that models are unable to re-

liably distinguish between conserving and non-conserving scenarios. The gap between average accuracy and strict pairwise evaluations suggests that models' success is driven largely by a bias toward quantity invariance rather than genuine reasoning about physical transformations. This finding further supports the conclusion that models fail to internalize structured physical reasoning and instead rely on brittle default strategies for quantity assessment.

**Dissociating Sources of Bias.** To dissociate the source of bias—whether it arises from visual features or textual priors—we conducted two control experiments on 62 VLMs that support both image and text-only inputs. First, we reran the same experiments using fully white, content-free images while keeping all text input constant (**Empty Image Control**). Second, we removed visual input entirely, presenting only text prompts (**Text Control**). We used the 7-frame condition for all comparisons to enable direct pairwise evaluation. Model responses were evaluated as if they were answering standard conservation tasks. If performance were driven purely by visual cues, models should operate at chance when visual content is removed. Conversely, systematic deviations from chance indicate reliance on textual biases—favoring either conservation (bias toward invariance) or non-conservation (bias toward perceptual change).

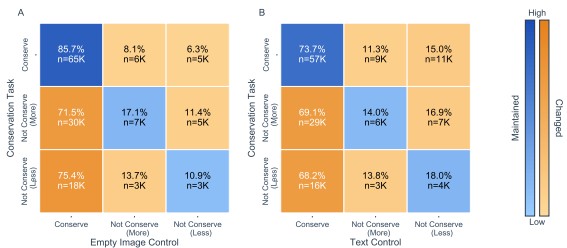

*Figure 3.* Transition matrices showing how model predictions shift when visual input is removed (**A.** Empty Image Control; **B.** Text-only Control). Each cell shows the proportion of samples transitioning from a prediction under full visual input (row) to a prediction under the control condition (column). The large values in column 1, rows 2–3 reveal a strong textual prior toward quantity invariance: models shift toward "Conserve" ∼70% of the time when visual content is removed. Paradoxically, models perform *worse* on actual conservation tasks with real visual content (∼60%) than without it (85.7%), indicating that visual content merely disrupts the invariance bias rather than enabling genuine transformation reasoning, producing the observed inverse performance across conserving and non-conserving conditions.

Figure 3 reveals striking patterns in how models respond when visual information is degraded or removed. In the Empty Image Control (Panel A), when the actual conservation task required a "Conserve" answer, 85.7% of responses remained "Conserve" under empty images. Critically, when the actual task required "Not Conserve" answers, models overwhelmingly switched to "Conserve" responses—71.5% for "More" scenarios and 75.4% for "Less" scenarios. The Text Control (Panel B) shows a similar but slightly attenu-

ated pattern: 73.7% maintain "Conserve" answers, while 69.1% and 68.2% of non-conserving scenarios shift to "Conserve" responses. To verify these findings are not an artifact of the forced three-choice format, we additionally re-ran these experiments with an added "I don't know" option on a subset of $n = 10$ models; the correlations remain robust (Empty Image: $r = 0.591$, $p < 0.0001$; Text Control: $r = 0.362$, $p < 0.0001$), confirming that textual biases persist regardless (see Appendix L). Further model-level details are provided in Appendix K.

These results reveal that textual priors strongly favor quantity invariance—a bias that is *correct* for conservation tasks but *incorrect* for non-conserving controls, explaining the inverse correlation we observe between the two task types. Notably, removing visual content while maintaining the visual modality (Empty Image) yields stronger conservation responses (85.7%) than removing the visual modality entirely (Text Control: 73.7%), suggesting that the presence of the visual channel amplifies textual biases even without meaningful visual information. Critically, however, models perform *worse* on actual conservation tasks with real visual content (average accuracy ∼60%) than with empty images (85.7%), indicating that visual content actively interferes with the textual prior rather than enabling transformation reasoning. This demonstrates that the core deficit lies in visual transformation reasoning: models cannot reliably extract and integrate transformation-relevant information from sequential visual evidence, leading them to incorrectly reject quantity invariance even when visual content should confirm it. The combination of strong textual priors and impaired visual processing accounts for both the moderate success on conservation tasks and the systematic inverse failures on non-conserving controls.

### 4.4. Different Prompting Strategies, Frame Numbers, and Sampling Methods

We further analyzed model performance across three experimental factors—prompt type, frame count, and frame sampling method—evaluated separately for Number & Length versus Volume & Size conservation tasks. To properly account for the hierarchical structure of our data (multiple observations nested within 112 models), we employed repeated-measures ANOVA with models as the unit of analysis. For each model, we first averaged accuracy across the irrelevant experimental conditions (e.g., when testing frame number effects, we averaged across all prompt types and extraction methods), then conducted repeated-measures ANOVA to test main effects, followed by Bonferroni-corrected pairwise comparisons for significant factors. This approach accounts for the dependency structure in our data while avoiding inflated Type I error from treating non-independent observations as separate trials. We highlight the main conclusions below.

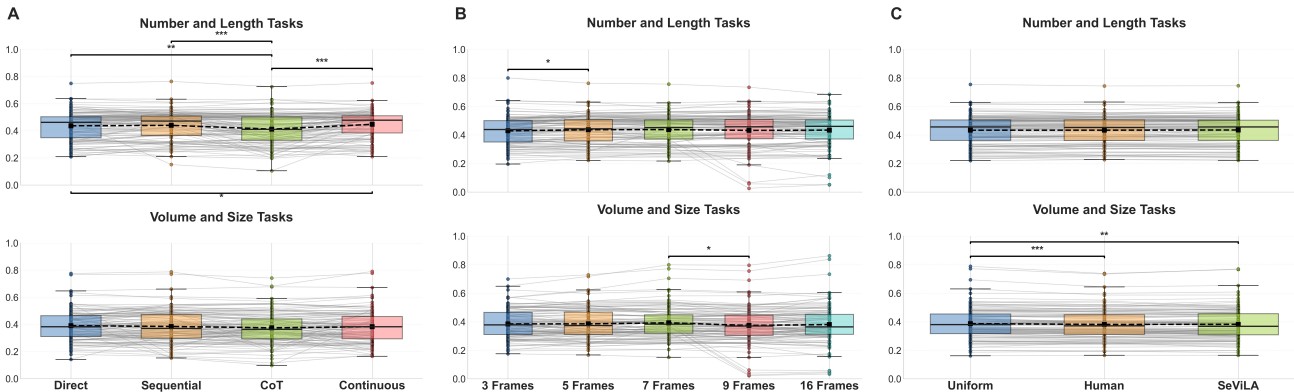

*Figure 4.* Model performance showing main effects by (A) prompt type, (B) number of frames, and (C) frame sampling method. Each panel averages across the other two factors from the full factorial design (4 prompts × 5 frame counts × 3 extraction methods). The dashed white line connects mean accuracies across conditions to facilitate visual comparison of trends.

**Continuous cues aid performance; CoT makes it worse (Figure 4A).** For Number & Length tasks, prompt type shows a highly significant main effect ($F(3, 333) = 18.28$, $p < 0.001$). Bonferroni-corrected pairwise comparisons reveal that CoT prompting performs significantly worse than all other prompt types: Continuous ($p < 0.001$), Sequential ($p < 0.001$), and Direct ($p = 0.0022$). Additionally, Continuous prompts—which explicitly frame transformations as continuous processes—significantly outperform Direct questions ($p = 0.0191$). These results indicate that conceptual cues emphasizing continuity can provide modest benefit for transformation-helpful tasks, while forcing step-by-step verbalization consistently impairs performance, likely by amplifying reliance on brittle heuristics. However, for Volume & Size tasks, prompt type shows no significant effect ($F(3, 333) = 2.00$, $p = 0.114$), suggesting that linguistic scaffolding provides no benefit when transformation reasoning demands are higher.

To further probe whether the failure reflects task misunderstanding rather than a deeper reasoning deficit, we evaluated two additional prompting conditions on a subset of 12 models. First, following Li et al. (2025a), we prepended an explicit conservation definition to each question. Conservation accuracy remains high while non-conserving accuracy remains near floor across all domains, consistent with the standard prompting results. Second, we evaluated caption-only and video+caption conditions using detailed textual descriptions generated by Qwen2.5-72B. Both conditions produce a striking domain-dependent reversal: Number & Length retain the invariance bias, while Volume & Size flip to a non-invariance bias—suggesting captions describing continuous transformations trigger a perceptual-change heuristic. Crucially, neither condition yields balanced performance across domains. Together, these results confirm that the failure persists even when full situational context is provided and is not attributable to task misunderstanding (see Appendix J).

**Temporal resolution shows no reliable benefit across task types (Figure 4B).** For Number & Length tasks, frame count shows no significant main effect ($F(4, 444) = 0.98$, $p = 0.416$), indicating that additional frames do not reliably improve performance even when transformation cues are helpful. For Volume & Size tasks, frame count shows a modest significant effect ($F(4, 444) = 2.66$, $p = 0.032$), but Bonferroni-corrected pairwise comparisons reveal only one significant difference: 7 frames outperform 9 frames ($p_{\text{corrected}} = 0.0329$). This lack of consistent improvement with increased temporal information demonstrates that current VLMs are unable to effectively integrate sequential visual evidence for transformation reasoning. Additional frames do not enable models to track continuous physical changes, even when such tracking is essential for task success.

**Frame extraction shows significant task-dependent effects (Figure 4C).** For Number & Length tasks, extraction method shows no significant effect ($F(2, 222) = 1.36$, $p = 0.258$), suggesting that different sampling methods perform comparably when transformation reasoning is helpful but not mandatory. However, for Volume & Size tasks, extraction method shows a highly significant effect ($F(2, 222) = 8.75$, $p = 0.0002$). Bonferroni-corrected pairwise comparisons reveal that uniform sampling significantly outperforms both human-selected ($p_{\text{corrected}} = 0.0006$) and SeViLA-selected frames ($p_{\text{corrected}} = 0.0014$), with no difference between the two curated methods ($p_{\text{corrected}} = 1.0$). For transformation-mandatory tasks (Volume & Size), curated frame selection may inadvertently emphasize misleading static features, suggesting that models are unable to leverage task-relevant visual information for reasoning, further demonstrating the lack of genuine physical transformation understanding.

**Failure patterns persist among top-performing models.** To assess whether the above null effects are driven by the

large number of low-performing models, we replicate all three analyses restricted to the top 15 models ranked by average accuracy. No significant effects of prompt type or frame count emerge even among the most capable models, and the extraction method effect on Volume & Size tasks is preserved. These results confirm that neither linguistic scaffolding nor increased temporal information facilitates transformation reasoning, even for the strongest current VLMs (see Appendix M).

### 4.5. Does Scaling of Model Size Help?

The advancement of LLMs has been closely tied to the empirical scaling law—predictable power-law improvements in performance with increased compute, parameters, and training data (Kaplan et al., 2020; Henighan et al., 2020; Zhai et al., 2022)—as well as emergence, the abrupt appearance of qualitatively new abilities as models grow larger (Wei et al., 2022; Aghajanyan et al., 2023; Bubeck et al., 2023; Berti et al., 2025). This raises a natural question: *Does the capacity to understand physical transformations and conservation similarly emerge with scale?*

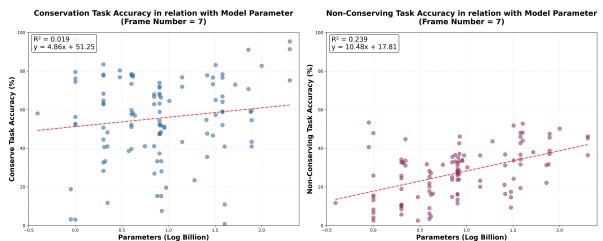

*Figure 5.* Conservation reasoning does not emerge with model scale. Model performance on (left) conservation tasks shows no relationship with parameter count ($R^2 = 0.019$), while (right) non-conserving task accuracy exhibits only modest scaling effects ($R^2 = 0.239$), both evaluated at 7-frame condition across 112 VLMs.

To this end, we examine performance versus model size (measured in log-scale parameters) across 112 VLMs ranging from 1B to 76B parameters, holding frame number constant at 7 to control for confounding effects of multi-frame processing capacity. We find strikingly divergent patterns (Figure 8). For conservation tasks, model size exhibits virtually no predictive power ($R^2 = 0.019$). In contrast, non-conserving task accuracy exhibits a moderate positive relationship with model size ($R^2 = 0.239$), indicating that larger models tend to perform better on non-conserving controls—yet even this relationship accounts for less than 24% of the variance. These results demonstrate that conservation reasoning improves minimally with scale in current VLMs; rather, scaling appears to amplify the existing bias toward non-conserving responses without enabling genuine transformation reasoning. To further test whether physically-oriented post-training can overcome these limitations, we additionally evaluated Cosmos Reason (Azzolini

et al., 2025), post-trained on physical common-sense and embodied reasoning data. Similar failure patterns persist, suggesting that the deficit is not remedied by physically-oriented post-training alone (see Appendix N).

### 4.6. Relationship to Other Physical Reasoning Benchmarks

ConservationBench is designed to isolate transformation-invariant reasoning under controlled conditions, whereas existing benchmarks embed physical understanding within richer contextual scenarios that introduce confounding factors; we view these as complementary. To substantiate this relationship empirically, we conducted a cross-benchmark correlation analysis using models available via VLMEvalKit (Duan et al., 2024), finding that ConservationBench correlates substantially with all nine benchmarks evaluated: BLINK (Fu et al., 2024), MLVU (Zhou et al., 2025), MMBench-Video (Fang et al., 2024), MME-RealWorld (Zhang et al., 2025), MMStar (Chen et al., 2024), NaturalBench (Li et al., 2024b), PhysBench (Chow et al., 2025a), RealWorldQA (xAI, 2024), and Video-MME (Fu et al., 2025a) (Figure 6). Notably, the correlation with Video-MME suggests that conservation reasoning and general video understanding share a common representational foundation, consistent with our hypothesis that the core bottleneck is the failure to integrate sequential visual evidence into stable object-state representations. Together, these results indicate that ConservationBench captures a general factor of visual physical understanding transferable to naturalistic settings, while its controlled design ensures observed failures can be specifically attributed to transformation reasoning deficits.

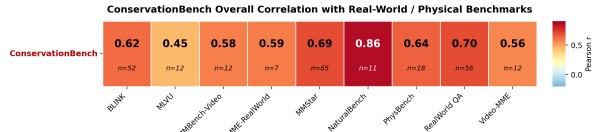

*Figure 6.* Cross-benchmark correlation between Conservation-Bench and nine physical reasoning and video understanding benchmarks, demonstrating that conservation reasoning captures a general factor of physical understanding.

### 4.7. Mechanistic Analysis of Failure Modes

Beyond behavioral characterization, we probe *why* models fail by analyzing the internal processing of Qwen2.5-VL-7B-Instruct on non-conserving (NC) trials, where the dominant error is confidently predicting "same" despite a genuine quantity change.

**Confidence Analysis.** We measure decision confidence via the log-probability margin between the top-1 and top-2 answer choices at the final decision token, comparing four prediction categories: *C_correct*, *NC_correct*, *NC_same_error*, and *NC_other_error* (Figure 16, Appendix O). Strikingly, *NC_same_errors* are substantially

*more* confident than *NC_correct* predictions, ruling out the interpretation that failures are uncertain fallbacks and instead indicating a strong committed prior toward quantity invariance. Meanwhile, *NC_correct* predictions are less confident than *C_correct*, suggesting that correct non-conserving judgments require overriding a default bias rather than reflecting principled reasoning.

**Attention Analysis.** We aggregate attention weights over visual tokens within each of the 7 input frames at the final decision token, normalizing across frames per layer to obtain a per-frame attention distribution (Figure 17, Appendix O). *NC_same_errors* place substantially more attention on Frame 1 (initial state) and less on later frames than *NC_correct* cases, particularly in late layers where decision-relevant processing occurs. This Frame-1 anchoring bias is absent for *NC_other_errors* and conserving-condition errors—it is not a generic property of incorrect predictions, but a pattern specific to the dominant failure mode.

Together, these analyses reveal a coherent failure mechanism: on non-conserving trials, the model anchors on the initial visual state and fails to update its representation after the transformation, yielding a confident but incorrect "same" response. This points to a fundamental bottleneck—current VLMs are predominantly built on static image encoders with weak temporal aggregation, making them effective at recognizing static visual features but inadequate for tracking object-state changes over time. Without mechanisms for maintaining and updating representations across frames, transformation reasoning is effectively reduced to static visual comparison. We note this analysis is preliminary on a single model; generalizability across model families and causal verification through targeted interventions are planned for future work.

## 5. Discussion

We introduce a cognitively grounded benchmark evaluating whether VLMs can reason about physical transformations through conservation tasks and non-conserving controls. Our contribution is not only empirical but diagnostic: through controlled experiments, we isolate a more fundamental limitation than low accuracy alone—models show systematic heuristic reversals, and control experiments reveal that textual priors dominate while visual evidence fails to correct them, suggesting the core deficit is not lack of knowledge but failure to extract and maintain object-state representations from visual input. Neither increased temporal resolution, targeted prompting, nor human-curated frame sampling induces robust transformation reasoning. These results expose a fundamental deficit in structured physical understanding and highlight critical challenges for developing grounded AI systems capable of systematic inference in dynamic environments. Beyond documenting these failures,

the tasks curated in this study can serve as enduring sanity checks for transformation reasoning, precisely because they expose failure modes that persist even as models advance on high-level physical reasoning benchmarks.

The failures documented here have direct implications for model design. Conservation reflects a foundational cognitive substrate scaffolding higher-level physical reasoning in humans, and models lacking such transformation-invariant reasoning pose risks for robust physical understanding in complex real-world scenarios (Li et al., 2025a; Cai et al., 2025). Our results suggest that current VLMs are predominantly built on static image encoders with weak temporal aggregation—sufficient for recognition or description, but inadequate for tracking quantity-preserving transformations over time. Critically, even fine-tuning, reasoning-oriented post-training, or in-distribution supervision is unlikely to resolve this underlying issue, as conservation judgments are visually intuitive and should not require task-specific training. Addressing these failures likely requires a different representational foundation: architectures that learn predictive, state-based visual abstractions rather than static semantic features (Zhang et al., 2024a; Luo et al., 2025a; Fu et al., 2025b). While our preliminary mechanistic analyses provide initial evidence for specific failure signatures, further mechanistic investigations are needed to precisely identify the representational constraints preventing current models from constructing transformation-invariant object representations.

## 6. Conclusion and Limitation

We introduce a cognitively grounded benchmark evaluating *conservation*—the principle that physical quantities remain invariant under transformations despite appearance changes. Current VLMs consistently fail to maintain transformation-invariant representations across dynamic scenes, indicating fundamental deficits in systematic physical reasoning that pose risks for embodied AI deployment. Our benchmark provides an enduring diagnostic test for transformation reasoning as the field advances.

We acknowledge several limitations. First, our evaluation focuses on four quantitative properties under controlled laboratory conditions; more complex scenarios involving occlusions, deformable objects, noisy observations, and ambiguous transformations are planned for future work. Second, while our preliminary mechanistic analyses provide initial evidence for specific failure signatures consistent with coarse-grained visual encoding bottlenecks, causal verification across model families remains for future investigation. Finally, whether conservation deficits impair downstream goal-directed tasks such as planning, tool use, or robotic manipulation remains an open empirical question.

## Impact Statement

This paper presents work whose goal is to advance the field of Machine Learning. There are many potential societal consequences of our work, none which we feel must be specifically highlighted here.

## Acknowledgments

This work was supported by a MiraclePlus Compute Grant to Growing AI Like A Child (https://growing-ai-like-a-child.github.io/)..

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

# A. Data Curation

**Curation and Quality Control.** *ConservationBench* was curated by three annotators with college-level training in cognitive science or computer science. Each video underwent two independent cross-review passes; items failing to meet design criteria were removed or revised.

**Data Acquisition.** All videos were captured under standardized recording conditions using a fixed camera setup, with consistent lighting and background held constant within each property category. Each transformation was carefully scripted to ensure visual clarity, reproducibility, and minimal ambiguity.

**Design Principles.** To ensure conceptual integrity and interdisciplinary rigor, we adopt three design criteria for each item: (i) *Discriminativeness*—tasks are constructed such that models lacking the targeted knowledge are systematically driven toward incorrect responses; (ii) *Minimal confounding*—instances are designed to minimize reliance on ancillary skills (e.g., object recognition); and (iii) *Minimal textual shortcuts*—tasks cannot be solved using textual cues alone and instead require genuine multimodal reasoning.

# B. Task Design

Table 2 presents paired descriptions of conservation tasks and their matched non-conserving controls across all four quantitative properties, with corresponding illustrations in Figure 2.

*Table 2.* Task descriptions for conservation and non-conserving control scenarios across four quantitative properties.

| Property | Conservation Task | Non-conserving Control |
|---|---|---|
| **Number** | Two rows of identical coins are presented in an initial configuration. One row is then spread apart without adding or removing any coins. | Two rows of identical coins are presented in an initial configuration. One row is then spread apart, with one coin added to said row. |
| **Length** | Two straws of identical length are shown in an initial configuration. One of the straws is then repositioned without altering its actual length. | Two straws of identical length are shown in an initial configuration. One of the straws is then repositioned while its actual length is altered (extendable straws are used). |
| **Volume** | A fixed volume of liquid is poured from one container into another of a different shape. Although the height changes significantly, the volume remains constant throughout the transformation. | A fixed volume of liquid is poured from one container into another of a different shape. A significant portion of water is left in the original container instead of being completely poured in. |
| **Size** | A lump of playdough is reshaped from one form (e.g., a ball) into another (e.g., a flattened disc). While the shape and surface features change, the total mass remains the same across both states. | A lump of playdough is reshaped from one form (e.g., a ball) into another (e.g., a flattened disc). A significant portion of playdough is left in the experimenter's hand without being integrated into the new shape. |

# C. Prompting Strategy

Reasoning about conservation often requires interpreting the transformation as a continuous process across the videos or sequence of frames. To examine how prompts influence temporal integration and transformation-based reasoning, we design four prompt types, each progressively enhancing the model's awareness of the underlying continuous process, as summarized in Table 3.

Together, these prompting strategies enable us to evaluate how different forms of linguistic scaffolding shape model engagement with visual dynamics. The "Sequential" and CoT prompts encourage frame-by-frame perception with step-by-step reasoning, directing attention to frame-wise visual evidence. In contrast, the "Continuous" prompt explicitly presents the multi-frame input as a continuous process, offering a conceptual cue to support conservation reasoning.

*Table 3.* Four different prompt formats used in our benchmark.

| Prompt Type | Prompt Example |
| --- | --- |
| **Direct Question** | Is the number of coins in the upper row the same as in the lower row in the final image? |
| **"Sequential" Prompt** | Please process the images below sequentially, and then answer: Is the number of coins in the upper row the same as in the lower row in the final image? |
| **CoT Prompt** | Please process the images below sequentially. First describe what happens across the images, then answer: Is the number of coins in the upper row the same as in the lower row in the final image? |
| **"Continuous" Prompt** | The above images represent a continuous process. Please answer: Is the number of coins in the upper row the same as in the lower row in the final image? |

## D. Example Input

To provide clarity on the exact format of inputs provided to models, we present a complete example task below, including both the visual frames and the full textual prompt.

**Task:** Conservation of Number (Conserving condition)

**Task Configuration:** This example demonstrates a Number conservation task using Uniform extraction method with 7 frames and Direct Question prompt format.

**Visual Input:** The model receives a sequence of frames extracted from the video in temporal order (Frame 1 through Frame 7), ensuring that the transformation process is presented chronologically without any frame order disruption. Figure 7 shows an example with 7 frames, where frames are sampled uniformly across the video timeline. The first frame shows the initial state (two rows of coins with equal numbers), intermediate frames capture the transformation process (spreading one row), and the final frame shows the end state (one row spread out while maintaining the same number of coins).

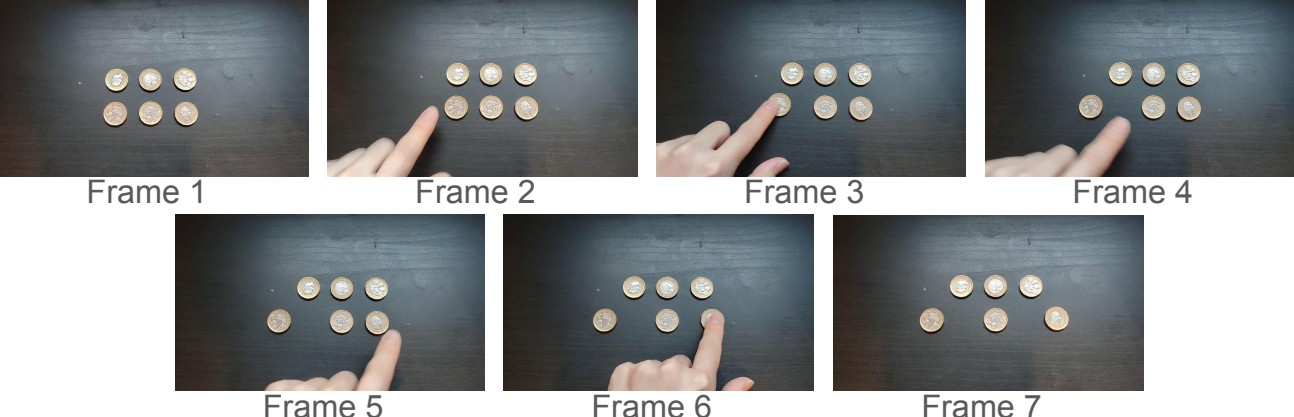

*Figure 7.* Example visual input: A sequence of 7 frames from a number conservation task, showing the initial state, transformation process, and final state.

**Textual Input:** Below is the structure of the prompt provided to the model (using the "Direct Question" format). The [Image] placeholders indicate where the corresponding frames from Figure 7 are embedded in the actual input:

```
Frame 1: [Image]
Frame 2: [Image]
Frame 3: [Image]
Frame 4: [Image]
Frame 5: [Image]
Frame 6: [Image]
```

```
Frame 7: [Image]
Is the number of coins in the upper row the same as in the lower row
in the final image?
Please choose one of the following options:
(A) No, the lower row has more coins.
(B) No, the upper row has more coins.
(C) Yes, they are the same.
```

**Ground Truth:** Option (C) - Yes, they are the same.

**Alternative Prompt Formats:** For other prompt types, the question is prefixed with additional instructions. For example, the "Sequential" format would begin with "Please process the images below sequentially, and then answer: [question]", while the CoT format would include "Please process the images below sequentially. First describe what happens across the images, then answer: [question]". See Table 3 for details on all four prompt formats.

## E. Model Inference

We evaluate 112 VLMs spanning diverse architectures, training regimes, and parameter scales, including mainstream proprietary models as well as advanced open-source models ranging from 1B to 76B parameters. Inference is conducted on a cluster equipped with 8× NVIDIA H100 (80 GB) GPUs. As a practical policy, models of 1–13B parameters typically run on a single GPU; 13–32B on two GPUs; 32–70B on four GPUs; and >70B on all eight GPUs.

To preserve fidelity and reproducibility, we adhere to configurations and reference implementations from the official codebases, avoiding unnecessary modifications. We build a scalable evaluation framework supporting parallel execution and compartmentalized environments. Inference jobs are distributed across GPUs via a dynamic scheduler that maximizes utilization and minimizes idle time. We additionally develop a lightweight modality-verification suite that prompts each model to summarize the media information it receives, and then the responses are checked by human reviewers to verify correct input routing and modality handling in our inference pipelines.

## F. Evaluation

Rule-based template matching degrades with complex model outputs, yielding elevated false positives/negatives and requiring continual template optimization to cover corner cases. LLM-based matching better identifies intended choices within free-form text but can hallucinate, especially when brief answers are embedded in extensive context. To balance these trade-offs, we introduce Hybrid Matching, which prioritizes deterministic template matching and, on failure, falls back to an ensemble of four LLM judges (Qwen2.5-72B-Instruct, Mixtral-8x7B-Instruct-v0.1, DeepSeek-R1-Distill-Llama-70B, and Llama-3.1-70B). The ensemble decision is accepted only if at least three three of four models return a consistent extraction; otherwise, the mapping is deemed unsuccessful. By coupling the precision of template extraction with the semantic flexibility of LLM adjudication, Hybrid Matching delivers more reliable mappings across diverse response styles.

# G. Counterbalancing Conditions

Complete counterbalancing parameters and their factorial combinations for all four quantitative property domains are provided in Table 4.

*Table 4.* Counterbalanced variations of task-irrelevant features. Each unique combination of parameter values yields 48 distinct task instances per domain.

| Domain | Parameter | Variations |
|---|---|---|
| Number | P1: Object Type | 2 variants (Uniform, Mixed) |
| | P2: Mapping Shift | 2 variants (Lower vs. Upper row moved) |
| | P3: Distance Spread | 2 variants (Near, Far) |
| | P4: Number of Objects | 6 variants (3–8 coins) |
| | Total combinations: | $2 \times 2 \times 2 \times 6 = 48$ |
| Length | P1: Object Type | 2 variants (Uniform, Mixed) |
| | P2: Mapping Shift | 2 variants (Lower vs. Upper straw moved) |
| | P3: Distance Moved | 2 variants (Near, Far) |
| | P4: Direction | 2 variants (Left, Right) |
| | P5: Transformation Action | 3 variants (Slide, Rotate, Vertical) |
| | Total combinations: | $2 \times 2 \times 2 \times 2 \times 3 = 48$ |
| Volume | P1: Liquid Color | 8 variants |
| | P2: Glass Transaction | 2 variants (Tall $\rightarrow$ Short, Short $\rightarrow$ Tall) |
| | P3: Liquid Volume | 3 variants (Small, Medium, Large) |
| | Total combinations: | $2 \times 8 \times 3 = 48$ |
| Size | P1: Object Color | 8 variants |
| | P2: Shape Transformation | 6 variants (Crossing Sphere, Cylinder, Plane) |
| | Total combinations: | $6 \times 8 = 48$ |

# H. Complete Model Results Aggregated Across Domains and Conditions

Complete performance metrics for all 112 evaluated VLMs, ranked by average accuracy, are provided in Tables 5 and 6.

*Table 5.* Complete Model Rankings by Average Accuracy (Ranks 1-56)

| Rank | Model | Conserve (%) | Non-Conserve (%) | Average (%) | Strict (%) |
|------|-------|--------------|------------------|-------------|------------|
| **0** | **Human Baseline** | **98.55** | **98.15** | **98.35** | **96.72** |
| 1 | gemini-2.5-pro | 94.33 | 43.88 | 69.11 | 41.20 |
| 2 | doubao-seed-1.6-vision | 82.99 | 46.42 | 64.71 | 38.22 |
| 3 | gpt-5 | 91.48 | 35.82 | 63.65 | 33.14 |
| 4 | claude-sonnet-4-5 | 75.43 | 45.39 | 60.41 | 37.23 |
| 5 | Qwen-3-VL-32B-Instruct | 64.57 | 52.06 | 58.31 | 31.15 |
| 6 | Qwen-2.5-VL-72B-Instruct | 89.69 | 21.98 | 55.83 | 17.32 |
| 7 | hunyuan-t1-vision | 74.05 | 36.05 | 55.05 | 26.80 |
| 8 | InternVL-3-38B-Instruct | 61.29 | 46.36 | 53.83 | 29.74 |
| 9 | InternVL-3.5-38B-Instruct | 75.49 | 31.80 | 53.64 | 19.32 |
| 10 | InternVL-3.5-38B-MPO | 72.75 | 33.98 | 53.36 | 20.85 |
| 11 | InternVL-3-38B | 57.47 | 48.42 | 52.95 | 29.41 |
| 12 | InternVL-2.5-78B | 60.71 | 44.43 | 52.57 | 19.27 |
| 13 | Qwen-2.5-VL-32B-Instruct | 81.82 | 22.83 | 52.33 | 16.66 |
| 14 | InternVL-3.5-8B | 72.53 | 29.93 | 51.23 | 15.62 |
| 15 | Qwen-2-VL-72B-Instruct | 69.44 | 31.52 | 50.48 | 15.24 |
| 16 | InternVL-3.5-8B-Instruct | 74.53 | 26.06 | 50.30 | 15.45 |
| 17 | InternVL-2.5-38B | 64.97 | 35.57 | 50.27 | 19.12 |
| 18 | InternVL-2.5-78B-MPO | 55.97 | 44.39 | 50.18 | 23.42 |
| 19 | InternVL-3.5-8B-MPO | 73.87 | 26.25 | 50.06 | 15.28 |
| 20 | InternVL-3.5-1B-MPO | 96.39 | 2.46 | 49.42 | 1.60 |
| 21 | InternVL-3.5-1B-Instruct | 93.45 | 3.45 | 48.45 | 1.96 |
| 22 | InternVL-2-2B | 77.56 | 18.13 | 47.85 | 8.16 |
| 23 | Ovis1.6-Gemma2-9B | 74.24 | 20.23 | 47.24 | 7.26 |
| 24 | InternVL-3-14B-Instruct | 74.86 | 19.24 | 47.05 | 8.94 |
| 25 | Qwen-3-VL-32B-Thinking | 61.44 | 32.44 | 46.94 | 20.23 |
| 26 | InternVL-3.5-30B-A3B-MPO | 75.13 | 17.29 | 46.21 | 9.15 |
| 27 | InternVL-3-14B | 69.07 | 22.90 | 45.99 | 11.00 |
| 28 | Qwen-2-VL-7B-Instruct | 66.58 | 25.14 | 45.86 | 6.20 |
| 29 | InternVL-3.5-30B-A3B-Instruct | 77.19 | 14.47 | 45.83 | 6.84 |
| 30 | InternVL-2.5-38B-MPO | 50.52 | 41.09 | 45.81 | 20.24 |
| 31 | InternVL-2-40B | 43.35 | 47.93 | 45.64 | 14.50 |
| 32 | Qwen-2.5-VL-3B-Instruct | 76.93 | 14.25 | 45.59 | 2.89 |
| 33 | InternVL-2.5-1B-MPO | 76.47 | 14.52 | 45.49 | 5.07 |
| 34 | InternVL-2-26B | 73.85 | 16.26 | 45.05 | 5.15 |
| 35 | InternVL-3-2B-Instruct | 81.51 | 7.93 | 44.72 | 1.50 |
| 36 | R1-Onevision-7B | 67.66 | 21.74 | 44.70 | 13.49 |
| 37 | InternVL-3.5-1B | 81.93 | 6.74 | 44.33 | 3.06 |
| 38 | Phi-3.5-vision-instruct | 71.53 | 17.12 | 44.32 | 5.35 |
| 39 | InternVL-3-2B | 76.01 | 12.05 | 44.03 | 2.75 |
| 40 | InternVL-2.5-2B | 59.59 | 28.25 | 43.92 | 10.89 |
| 41 | InternVL-2.5-4B-MPO | 77.03 | 10.63 | 43.83 | 6.55 |
| 42 | llava-onevision-qwen2-7b-si-hf | 58.76 | 27.93 | 43.34 | 3.40 |
| 43 | InternVL-2-8B | 74.09 | 12.07 | 43.08 | 4.12 |
| 44 | Qwen-2.5-VL-7B-Instruct | 64.12 | 21.55 | 42.83 | 7.39 |
| 45 | llava-next-interleave-qwen-7b-dpo | 50.13 | 35.20 | 42.66 | 6.17 |
| 46 | InternVL-2-Llama3-76B | 55.67 | 29.24 | 42.46 | 11.90 |
| 47 | Qwen-3-VL-8B-Instruct | 53.12 | 31.52 | 42.32 | 8.59 |
| 48 | InternVL-3-8B-Instruct | 70.95 | 13.61 | 42.28 | 7.97 |
| 49 | InternVL-2.5-1B | 74.86 | 9.41 | 42.14 | 3.69 |
| 50 | InternVL-3-9B-Instruct | 51.41 | 32.72 | 42.07 | 7.41 |
| 51 | Mini-InternVL-Chat-4B-V1-5 | 79.77 | 4.24 | 42.01 | 0.75 |
| 52 | InternVL-3.5-4B-Instruct | 57.07 | 26.51 | 41.79 | 8.13 |
| 53 | InternVL-3-9B | 50.46 | 32.88 | 41.67 | 7.90 |
| 54 | Qwen-2.5-Omni-3B | 80.22 | 3.06 | 41.64 | 1.48 |
| 55 | VLAA-Thinker-Qwen2VL-2B | 48.60 | 34.20 | 41.40 | 5.31 |
| 56 | InternVL-2-4B | 78.26 | 4.22 | 41.24 | 1.76 |

*Table 6.* Complete Model Rankings by Average Accuracy (Ranks 57–112, cont.)

| Rank | Model | Conserve (%) | Non-Conserve (%) | Average (%) | Strict (%) |
|------|-------|--------------|------------------|-------------|------------|
| 57 | InternVL-3-8B | 65.55 | 16.74 | 41.14 | 8.12 |
| 58 | InternVL-3.5-4B | 59.04 | 22.43 | 40.73 | 8.73 |
| 59 | Phi-3-vision-128k-instruct | 57.61 | 23.54 | 40.58 | 6.28 |
| 60 | InternVL-2.5-4B | 72.66 | 8.06 | 40.36 | 4.66 |
| 61 | Qwen-2.5-Omni-7B | 66.48 | 14.11 | 40.30 | 7.84 |
| 62 | VLAA-Thinker-Qwen2.5VL-7B | 54.77 | 25.56 | 40.16 | 10.56 |
| 63 | InternVL-2.5-26B | 47.66 | 32.42 | 40.04 | 9.60 |
| 64 | Qwen-2-VL-2B-Instruct | 44.73 | 34.68 | 39.70 | 4.33 |
| 65 | InternVL-3-78B-Instruct | 45.27 | 33.92 | 39.60 | 15.14 |
| 66 | InternVL-3-78B | 43.04 | 35.90 | 39.47 | 15.97 |
| 67 | VLAA-Thinker-Qwen2VL-7B | 52.68 | 25.69 | 39.18 | 7.84 |
| 68 | InternVL-3.5-14B-Instruct | 42.58 | 35.31 | 38.95 | 9.46 |
| 69 | InternVL-3.5-38B | 58.69 | 19.16 | 38.92 | 11.28 |
| 70 | InternVL-3.5-4B-MPO | 52.60 | 25.08 | 38.84 | 7.67 |
| 71 | Qwen-3-VL-2B-Instruct | 50.82 | 25.99 | 38.40 | 5.62 |
| 72 | llava-onevision-qwen2-7b-ov-hf | 47.35 | 29.06 | 38.21 | 2.07 |
| 73 | llava-onevision-qwen2-7b-ov-chat-hf | 47.64 | 28.40 | 38.02 | 2.01 |
| 74 | InternVL-Chat-V1-5 | 52.10 | 23.64 | 37.87 | 6.76 |
| 75 | InternVL-3.5-2B | 63.97 | 11.61 | 37.79 | 4.23 |
| 76 | InternVL-3.5-2B-Instruct | 62.85 | 12.32 | 37.58 | 4.46 |
| 77 | xgen-mm-phi3-mini-instruct-interleave-r-v1.5 | 57.61 | 17.35 | 37.48 | 2.33 |
| 78 | InternVL-3.5-2B-MPO | 61.43 | 12.76 | 37.10 | 4.31 |
| 79 | InternVL-2.5-26B-MPO | 34.80 | 38.07 | 36.44 | 9.70 |
| 80 | Mini-InternVL-Chat-2B-V1-5 | 42.32 | 29.75 | 36.03 | 2.67 |
| 81 | Phi-4-multimodal-instruct | 33.79 | 36.94 | 35.36 | 5.45 |
| 82 | InternVL-2.5-8B | 47.28 | 23.09 | 35.19 | 5.82 |
| 83 | Qwen-3-VL-4B-Instruct | 49.05 | 20.83 | 34.94 | 6.77 |
| 84 | Qwen-3-VL-8B-Thinking | 52.92 | 16.31 | 34.61 | 7.42 |
| 85 | InternVL3_5-GPT-OSS-20B-A4B-Preview | 57.01 | 11.61 | 34.31 | 4.64 |
| 86 | VLAA-Thinker-Qwen2.5VL-3B | 36.44 | 31.66 | 34.05 | 9.13 |
| 87 | Mantis-llava-7b | 31.46 | 36.21 | 33.83 | 4.23 |
| 88 | InternVL-2.5-8B-MPO | 37.53 | 29.13 | 33.33 | 7.42 |
| 89 | InternVL-2.5-2B-MPO | 32.20 | 34.24 | 33.22 | 4.92 |
| 90 | llava-v1.6-mistral-7b-hf | 33.33 | 32.85 | 33.09 | 0.00 |
| 91 | InternVL-3-1B | 52.17 | 13.98 | 33.07 | 0.47 |
| 92 | InternVL-3-1B-Instruct | 52.93 | 13.09 | 33.01 | 0.25 |
| 93 | InternVL-Chat-V1-1 | 23.54 | 42.34 | 32.94 | 2.45 |
| 94 | Mantis-8B-Idefics2 | 29.57 | 36.30 | 32.93 | 1.48 |
| 95 | llama3-llava-next-8b-hf | 32.44 | 32.47 | 32.45 | 0.00 |
| 96 | llava-onevision-qwen2-0.5b-si-hf | 19.06 | 41.86 | 30.46 | 1.10 |
| 97 | llava-onevision-qwen2-0.5b-ov-hf | 3.70 | 54.84 | 29.27 | 0.54 |
| 98 | Qwen-3-VL-4B-Thinking | 40.77 | 17.66 | 29.21 | 6.73 |
| 99 | Mantis-8B-clip-llama3 | 18.65 | 39.71 | 29.18 | 2.79 |
| 100 | Mantis-8B-siglip-llama3 | 12.33 | 43.78 | 28.05 | 1.47 |
| 101 | InternVL-3.5-30B-A3B | 47.53 | 8.47 | 28.00 | 4.92 |
| 102 | Xinyuan-VL-2B | 11.82 | 44.11 | 27.97 | 3.20 |
| 103 | Mantis-8B-Fuyu | 20.31 | 35.07 | 27.69 | 0.36 |
| 104 | InternVL-Chat-V1-2 | 11.11 | 43.85 | 27.48 | 2.93 |
| 105 | InternVL-Chat-V1-2-Plus | 0.71 | 51.53 | 26.12 | 0.28 |
| 106 | Mantis-bakllava-7b | 17.10 | 34.44 | 25.77 | 1.15 |
| 107 | InternVL-2-1B | 3.04 | 47.55 | 25.30 | 0.34 |
| 108 | JanusFlow-1.3B | 29.39 | 20.25 | 24.82 | 1.76 |
| 109 | Janus-Pro-7B | 28.66 | 20.89 | 24.78 | 2.74 |
| 110 | Janus-1.3B | 29.05 | 17.61 | 23.33 | 0.66 |
| 111 | Janus-Pro-1B | 19.51 | 23.62 | 21.57 | 1.42 |
| 112 | Qwen-3-VL-2B-Thinking | 30.10 | 9.52 | 19.81 | 2.51 |

# I. Combined Model Performance By Quantitative properties

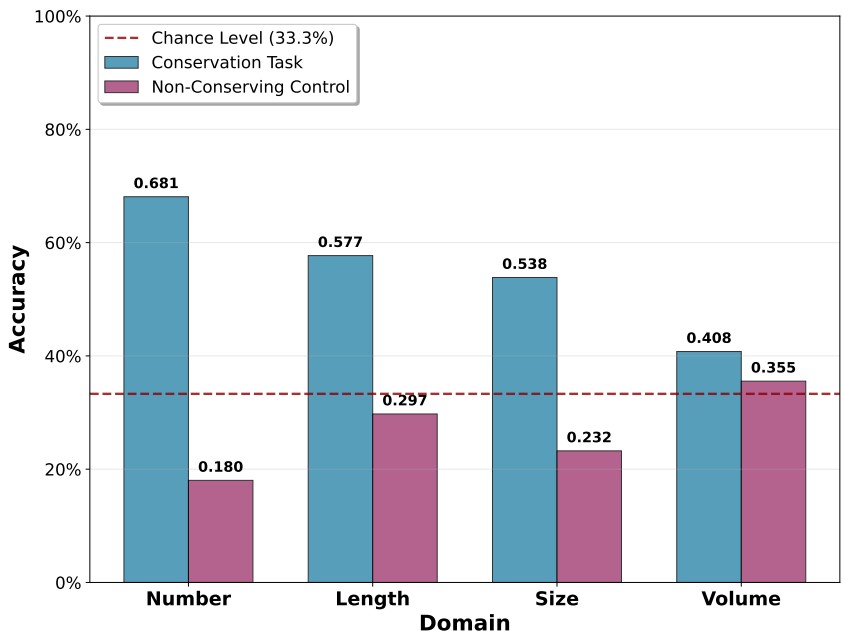

*Figure 8.* Average model performance by quantitative domains. Models consistently perform worse on non-conserving controls compared to conservation tasks.

## J. Additional Prompting and Input Modality Controls

**Concept Explanation Prompting.** We evaluated 12 models with explicit conservation definitions provided before each question: "*This question is about conservation, which is the understanding that physical quantities remain invariant under transformation despite changes in appearance. Now, answer:*" under the 7-frame uniform sampling setting. Results (Figure 9) show that conservation accuracy remains high (Number: 0.863, Length: 0.795, Volume: 0.681, Size: 0.742) while non-conserving accuracy remains near floor (0.014/0.053/0.128/0.046), consistent with performance without concept-based prompting. This demonstrates that providing explicit conceptual scaffolding does not enable models to flexibly resolve conservation tasks.

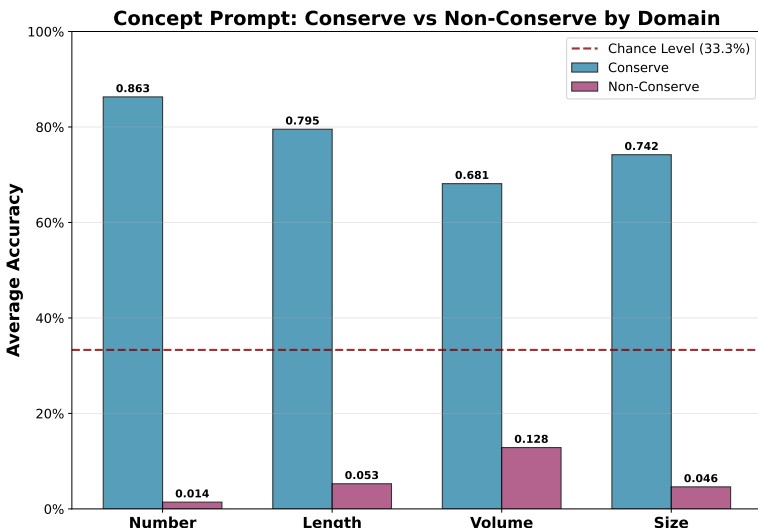

*Figure 9.* Model performance under concept explanation prompting across four quantitative properties.

**Caption-only and Video+Caption Controls.** We evaluated 12 models under two additional conditions: caption-only (text description of the video without visual input) and video+caption (video frames plus text description). Captions were generated by Qwen2.5-VL-72B without answer-leaking instructions and verified by human reviewers. Results (Figures 10 and 11) reveal domain-dependent variations: Number and Length retain the invariance bias (Conserve: ∼0.78/0.85; Non-Conserve: ∼0.06/0.11), while Volume and Size flip to a non-invariance bias (Conserve: 0.014/0.044; Non-Conserve: 0.798/0.684), suggesting that captions describing continuous transformations trigger a perceptual-change heuristic. Critically, neither condition approaches balanced performance across domains, demonstrating that the failure persists even when full situational context is provided in text—the deficit is in physical reasoning itself, not visual processing alone.

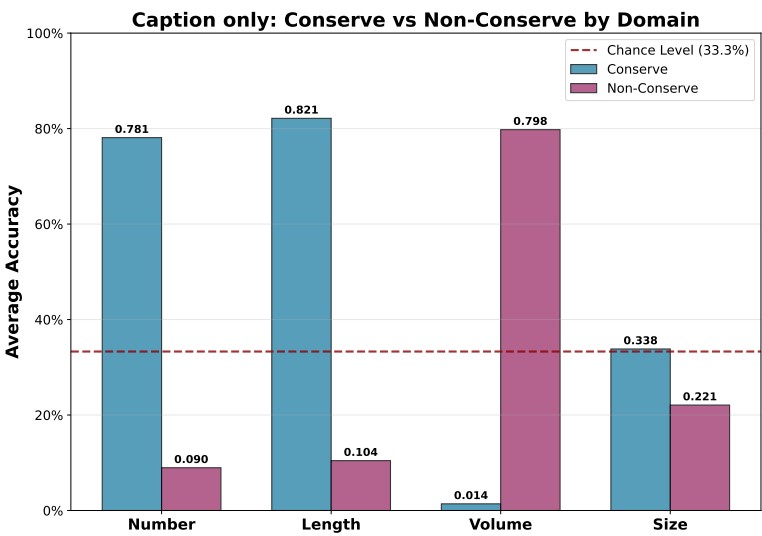

*Figure 10.* Model performance under caption-only conditions across four quantitative properties.

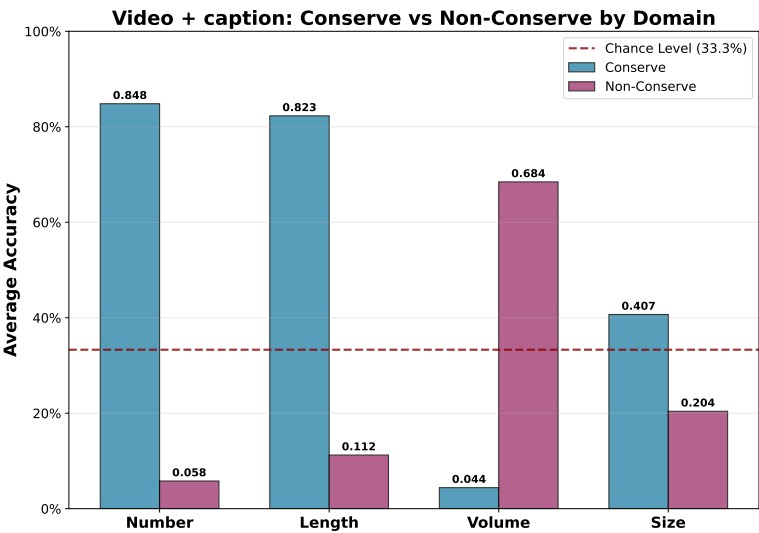

*Figure 11.* Model performance under video+caption conditions across four quantitative properties.

# K. Model response under empty image and text control

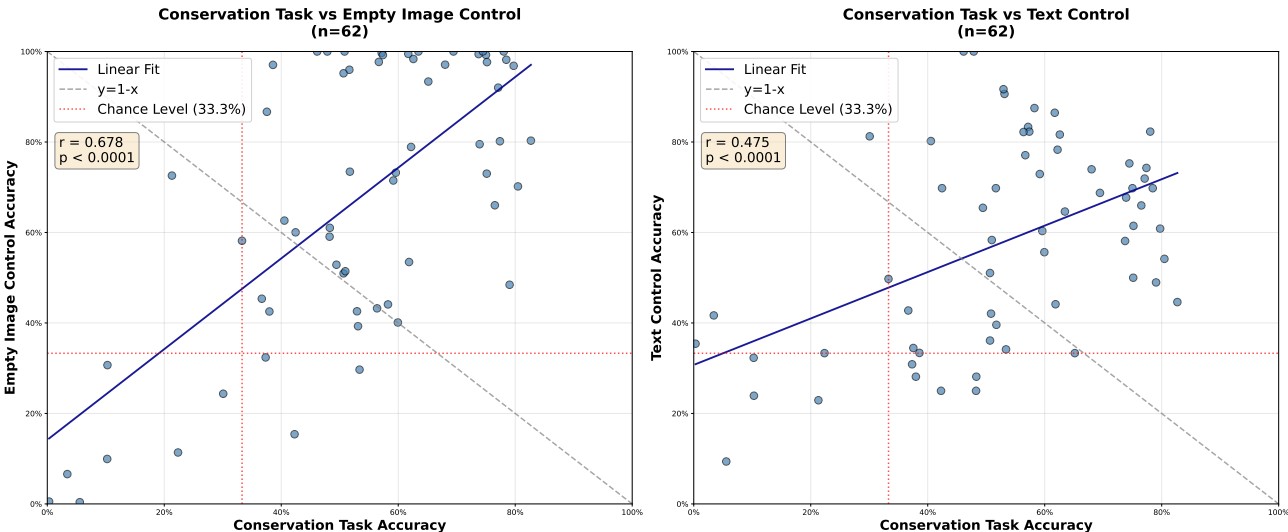

*Figure 12.* Model-level correlations between conservation task performance and control condition biases. (Left) Conservation task accuracy versus Empty Image Control accuracy shows strong positive correlation ($r = 0.578$, $p < 0.0001$), indicating models performing better on conservation tasks exhibit stronger textual priors favoring quantity invariance when visual content is removed. (Right) Conservation task accuracy versus Text Control accuracy shows similar but slightly weaker correlation ($r = 0.475$, $p < 0.0001$), with both patterns demonstrating that success on conservation tasks is driven primarily by textual biases rather than visual transformation reasoning. Evaluated on 62 VLMs supporting both empty image and text-only inputs under 7-frame conditions.

# L. Empty Image and Text Control with "I Don't Know" Option

We re-ran the Empty Image and Text Control experiments on a subset of $n = 10$ models with an added "I don't know" response option to test whether the observed textual biases persist when models can abstain from answering. Results are shown in Figure 13. The positive correlation between conservation task accuracy and control condition accuracy remains robust (Empty Image: $r = 0.591$, $p < 0.0001$; Text Control: $r = 0.362$, $p < 0.0001$), consistent with our original results ($n = 62$: $r = 0.678$ and $r = 0.475$). This confirms that both visual and textual biases are robust even when models have the option to withhold a judgment.

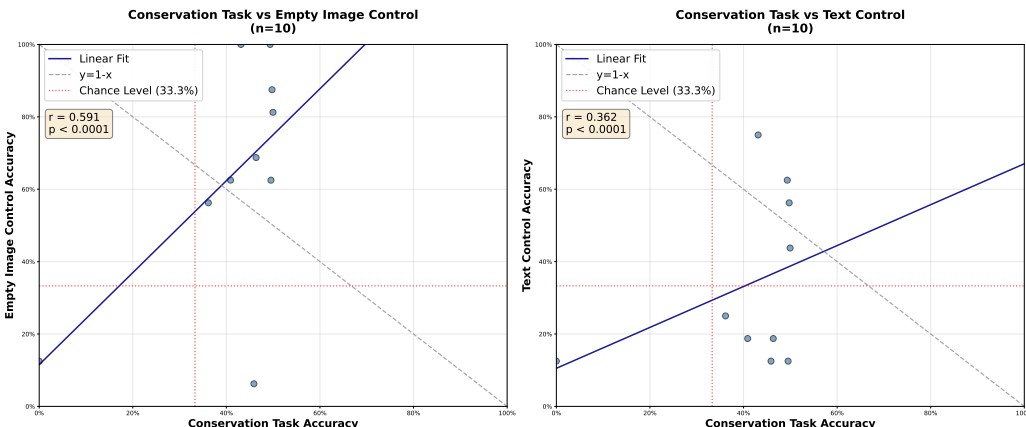

*Figure 13.* Correlation between conservation task accuracy and control condition accuracy with an added "I don't know" response option ($n = 10$ models).

## M. Prompting, Frame Count, and Extraction Method Analysis on Top-15 Models

To assess whether the null effects of prompt type and frame count reflect the inclusion of many low-performing models, we re-ran the Section 4.4 analyses restricted to the top 15 models ranked by average accuracy. Results are shown in Figure 14. No significant effects of prompt type or frame count are found for either task category among top models. The extraction method effect observed in the full sample is preserved—with SeViLA leading to significantly worse performance on Volume & Size tasks ($p < 0.01$)—further confirming that neither linguistic scaffolding nor increased temporal information facilitates transformation reasoning even among the most capable VLMs.

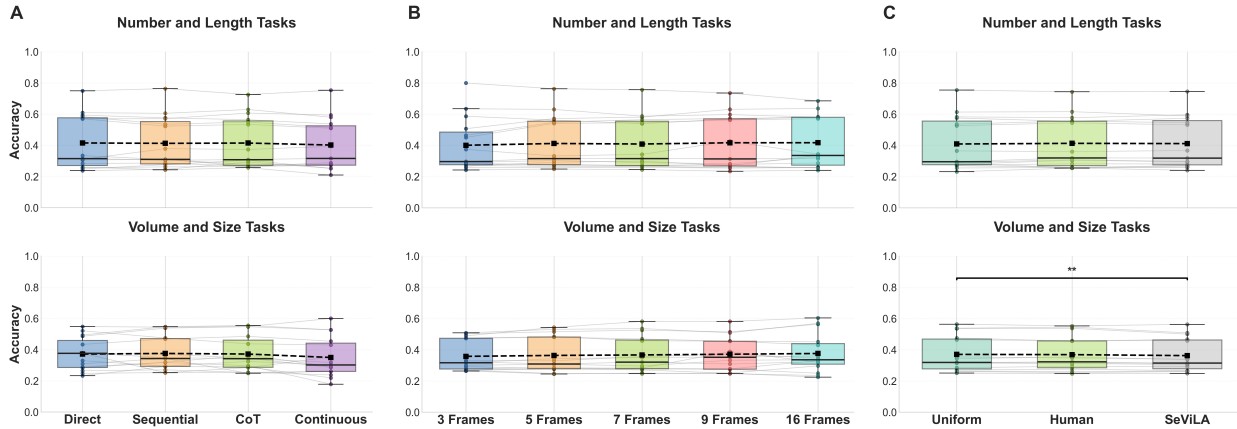

*Figure 14.* Prompt type, frame count, and extraction method effects restricted to the top 15 models by average accuracy.

## N. Evaluation of Cosmos Reason

We evaluated Cosmos Reason (Azzolini et al., 2025), a model post-trained with physical common-sense and embodied reasoning data, to assess whether physically-oriented post-training can overcome the failure modes documented in ConservationBench. Results are shown in Figure 15. Cosmos Reason exhibits a similar trend to all other models: near-zero non-conserving accuracy on Number (0.003) and Length (0.045), and below-chance performance on Volume (0.229) and Size (0.108). The characteristic asymmetry between conservation and non-conserving task performance persists, suggesting that post-training on physical common-sense data does not resolve the underlying representational limitations.

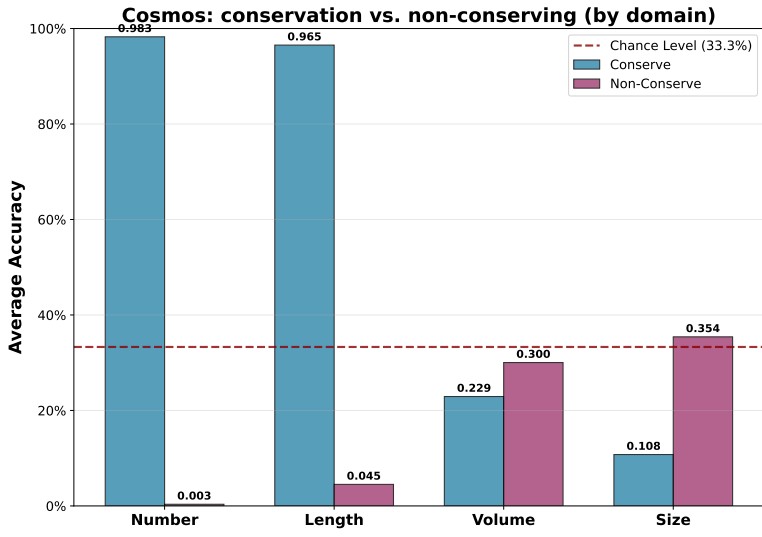

*Figure 15.* Performance of Cosmos Reason on ConservationBench across four quantitative properties.

# O. Mechanistic Analysis: Confidence and Attention

**Confidence Analysis.** Figure 16 shows decision confidence (log-probability margin between top-1 and top-2 answer choices) for four prediction categories on Qwen2.5-VL-7B-Instruct. *NC_same_errors* are substantially more confident than *NC_correct* predictions, while *NC_correct* predictions are less confident than *C_correct*, consistent with the interpretation that correct non-conserving judgments require overriding a default invariance prior.

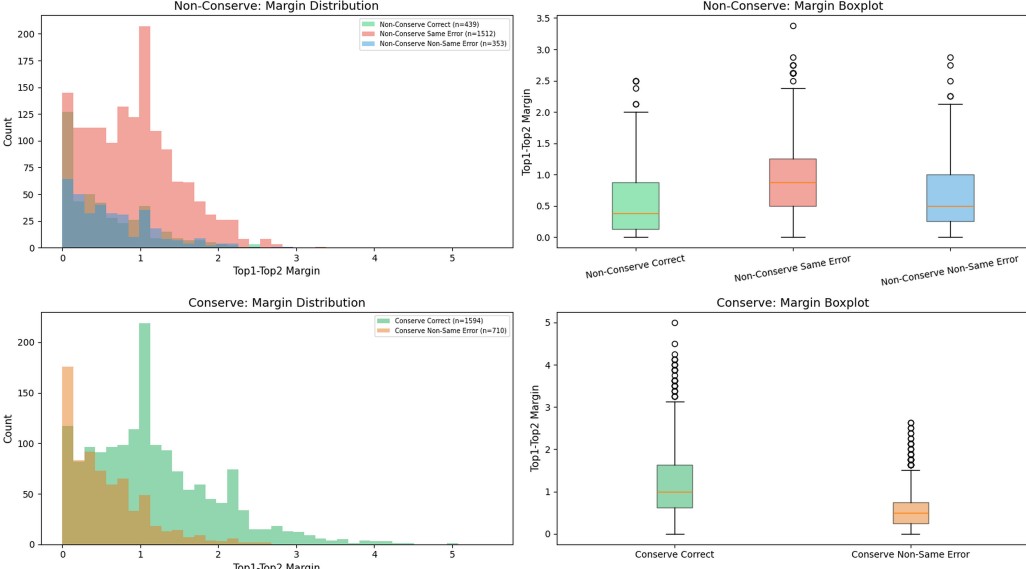

*Figure 16.* Decision confidence (log-probability margin between top-1 and top-2 answer choices) for four prediction categories on Qwen2.5-VL-7B-Instruct. *NC_same_error*: incorrect "same" prediction on non-conserving trials; *NC_correct*: correct prediction on non-conserving trials; *C_correct*: correct prediction on conserving trials; *NC_other_error*: other incorrect predictions on non-conserving trials.

**Attention Analysis.** Figure 17 shows layer-wise frame-attention heatmaps obtained by aggregating attention weights over visual tokens within each of the 7 input frames at the final decision token, normalized across frames per layer. The top rows show non-conserving (NC) trials across three prediction categories with pairwise difference maps (Δ); the bottom row shows conserving (C) trials. The key pattern is visible in the Δ NC_same_error − NC_correct map: NC_same_errors place disproportionately more attention on Frame 1 and less on later frames in late layers (layers 20–27), while this anchoring bias is absent in all other difference maps, confirming it is specific to the dominant failure mode.

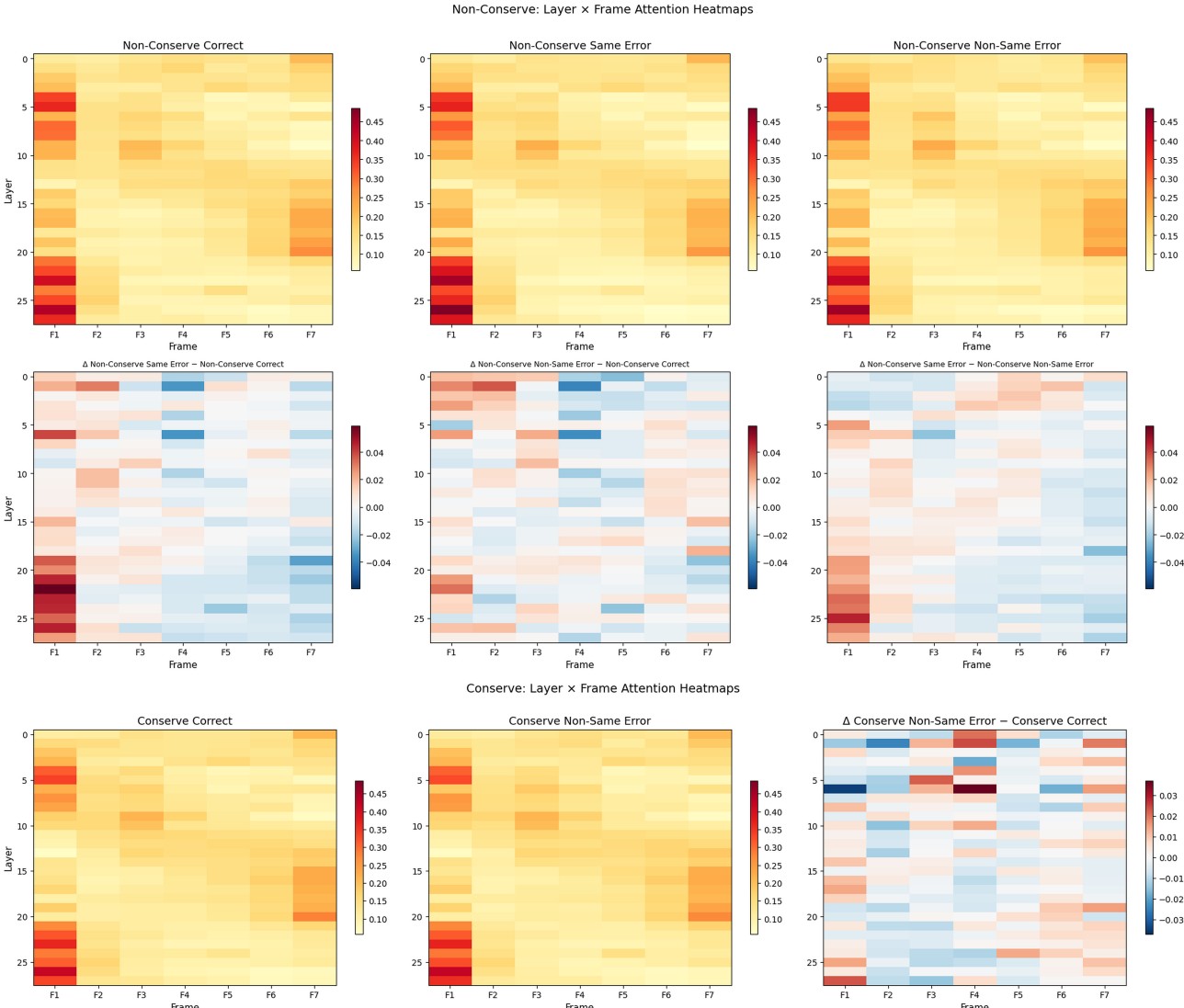

*Figure 17.* Layer-wise frame-attention heatmaps for Qwen2.5-VL-7B-Instruct. **Top:** Non-conserving (NC) trials showing absolute attention distributions for NC_correct, NC_same_error, and NC_non_same_error (top row), alongside pairwise difference maps Δ (bottom row). **Bottom:** Conserving (C) trials showing C_correct and C_non_same_error with their difference map. The key finding is visible in Δ NC_same_error − NC_correct: NC_same_errors place disproportionately more attention on Frame 1 (initial state) in late layers, a pattern absent in all other difference maps.

## P. Qualitative Analysis of Model Reasoning Traces

To complement our quantitative findings, we present qualitative examples of model reasoning traces on ConservationBench tasks. Figure 18 shows a representative failure case from CLAUDE-SONNET-4-5 on a non-conserving volume task. The model carefully compares liquid levels across frames and concludes that the quantities are equivalent. Although this response captures perceptual similarity, it fails to represent the transformation as a continuous physical process in which partial liquid transfer should dominate any static comparison of apparent levels. Rather than reasoning about whether the transfer is complete or partial, the model reduces the task to frame-by-frame perceptual matching. This is precisely the failure mode we reveal: models default to static visual matching instead of constructing transformation-invariant representations of the physical dynamics.

> "Let me analyze the images carefully to compare the amount of liquid in the left glass in the first image with the right glass in the final image.
>
> Image 1 (First image): Shows two glasses on a surface. The left glass contains an amber/orange colored liquid (appears to be whiskey or similar). The liquid level appears to be approximately at the middle of the glass, maybe slightly higher. The right glass is empty with just clear liquid at the bottom.
>
> Image 7 (Final image): Shows two glasses on a surface. Both glasses now contain amber/orange colored liquid. The left glass appears to have the liquid at a similar level to the first image. The right glass also has liquid, and it appears to be at approximately the same level as the left glass.
>
> Looking more carefully:
>
> - In Image 1: The left glass has a significant amount of amber liquid (roughly halfway up or slightly more)
>
> - In Image 7: The right glass appears to have a similar amount of liquid as the left glass in Image 1
>
> The sequence shows what appears to be a liquid transfer process where the liquid from the left glass is being poured into the right glass. By the final image, both glasses appear to have roughly equal amounts of the amber liquid, suggesting the original liquid was divided between them.
>
> So comparing:
>
> - Left glass in Image 1: Has amber liquid at approximately a certain level (looks like it fills about 1/3 to 1/2 of the glass height)
>
> - Right glass in Image 7: Has amber liquid at what appears to be a similar level
>
> They look approximately the same to me. The answer should be B - Yes, they are the same."

*Figure 18.* Example failure case from CLAUDE-SONNET-4-5 on a non-conserving volume task. The model performs frame-by-frame perceptual matching rather than reasoning about the transformation as a continuous physical process.

