# OpenReview forum: "Vision Language Models Cannot Reason About Physical Transformation"
_ICML.cc/2026/Conference — ICML 2026 regular_

### Official Review · Reviewer_mMyh · 2026-03-11

**Soundness:** 2
**Presentation:** 4
**Significance:** 3
**Originality:** 3
**Overall Recommendation:** 4
**Confidence:** 4

**Summary:**

This work investigates how a wide range of VLMs perform on simple reasoning about physical transformation -- whether target objects' quantity or size changed throughout a particular visual event. The evaluation consists of recorded videos of simple object and manipulations. The benchmark consists of balanced-design of change vs. no change visual events. The results show that, aside from a few top models, many VLMs fail to systematically answer correctly, and exhibit strong influence from textual priors on invariance.

**Compliance With Llm Reviewing Policy:**

Affirmed.

**Final Justification:**

I'm keeping my positive recommendation.

**Key Questions For Authors:**

- Including some concrete success/failure cases would enhance the paper.
- Have the authors ensured that all trials are answerable across all conditions?
- Are the authors planning on open-sourcing the benchmark?

**Limitations:**

The authors have adequately discussed limitations of their work.

**Strengths And Weaknesses:**

Strength:
- I find the video events in the benchmark quite interesting, especially with the matching control video for each test video and the intention to counterbalance task-irrelevant factors such as object types and spatial layout
- This work performed quite a comprehensive investigation w.r.t. frame selection and image ablations, accounting for textual prior/bias
- A wide range of VLMs are studied, which is quite an applaudable effort

Weakness:
- I find the title over-claiming -- even though models are far from humans, top models clearly perform above chance even in the strict-rating setting.
- I think it makes more sense to study the effect of prompt types of frame count only on successful models to understand their effect, grouping all models together may wash out these effects with the sheer amount of not-so-good models
- It's unclear how much reasoning capabilities are used in testing closed models, or if a bit more prompt engineering could get the models to perform more robustly.
- I'm concerned that some of the trials are not answerable -- e.g. with 3-frame uniform sampling, it may not capture the key event at all.

---

> ### Author Rebuttal · Authors · 2026-03-31
>
> W1: I find the title over-claiming -- even though models are far from humans, top models clearly perform above chance even in the strict-rating setting.
>
> > Thanks for the advice! While we understand that top models perform moderately above chance, we argue that Conservation is a **foundational cognitive capacity that scaffolds higher-level physical reasoning** (Piaget, 1965; Baillargeon & Carey, 2012) where **humans achieve near-perfect accuracy (~98%)**, as confirmed by our human baseline. Moderately above-chance performance for such an ability is vastly insufficient and likely doesn't constitute meaningful backbone for upstream capabilities, particularly given that strict pairwise evaluation reveals 73.2% of models fall below chance (<10% strict accuracy), and only 3 of 112 exceed it. Above-chance average accuracy is largely explained by a default invariance heuristic that is fortuitously correct for conservation tasks but wrong for matched non-conserving controls (r = −0.510, p < 0.0001). We will refine the title to incorporate this suggestion for better comprehension.
>
> W2: I think it makes more sense to study the effect of prompt types of frame count only on successful models to understand their effect.
>
> > Thanks for the suggestion. We re-ran the Section 4.4 analyses and only included the **top 15 models** on both conserve and non-conserve conditions. The results are shown in **https://imgur.com/a/zWIllSY**: no significant effects of prompt type or frame count are found for either task category, and the extraction method effect observed in the full sample is preserved---with SeViLA leading to significantly worse performance on Volume & Size tasks (p<0.01). This further confirms that neither linguistic scaffolding nor increased temporal information facilitates transformation reasoning, even among the most capable models.
>
> W3: It's unclear how much reasoning capabilities are used in testing closed models, or if a bit more prompt engineering could get the models to perform more robustly.
>
> > We thank the reviewer for this question. Our benchmark already systematically evaluates **four prompt types**---including Chain-of-Thought, which explicitly elicits step-by-step reasoning---and finds no reliable benefit (Section 4.4). Furthermore, as shown in our response to Reviewer qGb8 (W2), concept explanation prompting providing explicit conservation definitions also fails to improve performance. Fundamentally, **conservation judgments are visually intuitive tasks that humans solve effortlessly without specialized prompting**. Our benchmark is designed to assess basic visual understanding, not prompt-engineering proficiency.  If robust performance depends on such task-specific interventions, this suggests that the underlying visual representation remains insufficient. We will clarify this perspective in the revision.
>
> W4 & Q2: I'm concerned that some of the trials are not answerable -- e.g. with 3-frame uniform sampling, it may not capture the key event at all.
>
> > Thanks for raising this question. The 3-frame setting is intentionally designed as a minimal temporal-resolution condition and **was verified by our data curator and validator to preserve the information necessary for the task**. Furthermore, the human baseline shows that participants achieve near-perfect performance under this condition, also indicating that these trials provide sufficient contextual information and are answerable in principle.
>
> Q1: Including some concrete success/failure cases would enhance the paper.
>
> > Thanks for the suggestion. We have conducted a qualitative analysis on the reasoning traces of models' responses, which will be added to the revision. An example failure case is shown in **https://imgur.com/a/CTDSRg4**. We refer to W3 of Reviewer 1fAq for detailed discussion. We will add more representative success/failure cases in Appendix D in the final revision.
>
> Q3: Are the authors planning on open-sourcing the benchmark?
>
> > Yes, we plan to open-source everything, including data, raw evaluation output, and results on GitHub and HuggingFace. We will also open-source the evaluation infra which allows us to evaluate over 100 VLMs.

---

> > ### Author Rebuttal · Reviewer_mMyh · 2026-04-03
> >
> > I thank the authors for following up on these points, and appreciate the additional results/evidence. I remain hesitant to the strong title, by my concerns are mostly addressed. I intend to keep my positive recommendation.

---

> > > ### Author Response · Authors · 2026-04-03
> > >
> > > Thank you for your feedback! We are glad that our rebuttal addressed your concerns. We will adjust the title following your suggestion. We sincerely appreciate your positive assessment, and if you feel that the revised manuscript merits it, we would be grateful if your final evaluation could reflect this updated view.

---

### Official Review · Reviewer_LfSu · 2026-03-11

**Soundness:** 3
**Presentation:** 3
**Significance:** 2
**Originality:** 3
**Overall Recommendation:** 4
**Confidence:** 4

**Summary:**

The paper introduces a benchmark for evaluating how well vision language models can track conservation of physical properties.

**Compliance With Llm Reviewing Policy:**

Affirmed.

**Final Justification:**

The paper reflects an interesting addition to the growing body of literature on VLMs deficiencies. The authors did a good job of explaining the parts left unclear, and I am sure the updated version of the paper will be more clear and accessible, and I am glad to see all reviewers leaning toward acceptance of this paper.

**Key Questions For Authors:**

**Questions**
- I think you should offer more details on the setup for the human experiment. I could not find any detail on how many human subjects were tested or excluded, what kind of questions you asked them. Where you tested them?
- The result in Figure 3 is a bit puzzling to me. If you gave them models an empty Image but you asked them the same question, shouldn't the probabilities for all conditions be the same? I mean that regardless of whether the video actually shows a non-conserving or a conserving event, since the model just sees white frames should it not say conserving or non-conserving equally as often? Why is it that it still has a higher probability for saying non-conserving for actually non-conserving videos, when there is no visual information? Maybe I have missed something but this is very odd.
- What is the dotted line in Figure 4? I think it's visually quite distracting and I'm not sure it adds much.


**Minor comments**
- In the abstract in line 20 you write: "We generate 23,040 questions across 112 VLMs". This should probably read We evaluate 23,040 questions or we generate 23,040 questions and evaluate them across 112 VLMs or so?
- Also, in line 25: "yet models perform worse with visual content". I know what that means after having read the paper, but I think this could be written more clearly in the abstract.
- In line 140 on the right side, not sure counterfactual is the right word here? I would write "we create matched non-conserving videos" instead.
- In the caption to Figure 2, I think you should add something on what the strict calculation is. It takes quite a while in the text after until it is explained. I think this would improve readability.

**Limitations:**

The authors mention this themselves, but the paper identifies a problem in current vision language models without providing a mechanistic understanding of the problem or an approach for how it could be solved. As such, I think the impact is limited. We know that these models struggle with basic visual processing. Ideally, the paper would offer approaches for how it could be overcome, or a theoretical perspective that confidently explains why these limitations appear.

**Strengths And Weaknesses:**

- **Soundness**: The methodology is sound. I appreciate the pairing of conserving and non-conserving stimuli, as well as the ablations for prompting and number of frames, and the inclusion of a human baseline.
- **Presentation**: The paper is well written and well structured.
- **Significance**: The scope of the investigation is limited, it focuses only on conservation of physical properties in videos. However, I do think there is value in understanding these low-level perceptual abilities in vision language models.
- **Originality**: The work provides new insight into specific perceptual abilities of vision language models.

---

> ### Author Rebuttal · Authors · 2026-03-31
>
> Q1: I think you should offer more details on the setup for the human experiment.
>
> > Thanks for the question. We recruited 7 participants online through a professional data-collection platform with standardized quality-control procedures. One participant was excluded because their performance on a consecutive block of questions was at chance level, indicating inattention, leaving 6 participants for analysis. As described in Section 4.2, participants were evaluated on a counterbalanced subset of 864 questions (one of every eight task configurations for each quantitative property, balanced across conservation and non-conserving conditions). They received the same visual and textual stimuli as the VLMs through a human-friendly interface and selected answers directly, without any LLM-based answer parsing. We will update the paper with these details.
>
> Q2: The result in Figure 3 is a bit puzzling to me.
>
> > Thanks for raising this concern. It's reasonable to expect that model **should** output the same probabilities for all conditions. However, MLLMs are inevitably **biased** by various factors such as training data, language prior, etc. [1], which exactly leads to the motivation of our experiment: to dissociate the sources of bias. As shown in Figure 3, we show that without the visual information, models are biased strongly towards quantity invariance, explaining the inverse correlation we observe between conservation and non-conserving task performance.
> >
> > [1] Zhang, YiFan, et al. "Debiasing multimodal large language models via penalization of language priors." Proceedings of the 33rd ACM International Conference on Multimedia. 2025.
>
> Q3: What is the dotted line in Figure 4? I think it's visually quite distracting and I'm not sure it adds much.
>
> > Thanks for the question. To clearly, the bold black point on each box is the mean accuracy averaged across all sampels in the same condition. The dashed line links these points to better illustrate a comparison between conditions. We will add this clarification to the figure caption.
>
> L1: The paper identifies a problem in current vision language models without providing a mechanistic understanding of the problem or an approach for how it could be solved.
> > We agree that documenting a failure alone is not sufficient, and our goal is not merely to show that current VLMs underperform on yet another benchmark. Rather, we use this task to expose a more fundamental limitation in the current VLM paradigm: these models do not appear to build stable visual representations of objects and their state changes under transformation.
> >
> > **Diagnostic Experiments** Our contribution is therefore not only empirical, but diagnostic. Through controlled exps, we isolate the failure mode more precisely than prior work. The issue is not simply low accuracy. Models show systematic heuristic reversals, and empty-image / text-only controls further show that textual priors often dominate while visual evidence fails to correct them. This suggests that the core limitation is not lack of knowledge, but failure to reliably extract and maintain the relevant object-state representation from visual input.
> >
> > **Model Design Implications** We believe this has direct implications for model design. Current VLMs are largely built on static image encoders and weak temporal aggregation, which may be sufficient for recognition or description, but are not enough for tracking quantity-preserving transformations over time. Even if fine-tuning, reasoning-oriented post-training, or narrow in-distribution supervision improves benchmark performance, such gains would not resolve the underlying issue. Conservation-like judgments are visually intuitive and should not require task-specific hacks or specialized exposure to near-identical training examples. Our results instead suggest that solving this class of failures likely requires a different representational foundation: models that learn predictive, state-based visual abstractions rather than only static semantic features. Approaches based on predictive visual modeling, such as JEPA-style objectives, explicit latent state tracking, or world-model-like architectures, may provide a more principled path forward. We will revise the discussion to make these architectural implications clearer.
> >
> > **Internal Mechanisms Analysis** We have added a mechanistic analysis on Qwen2.5-VL-7B-Instruct, providing a clearer hypothesis for the failure mode. The confidence-margin analysis (**https://imgur.com/a/LiRw1Gm**) and the layer-wise frame-attention heatmaps (**https://imgur.com/a/uOGZmtM**) suggest that the model anchors excessively to the initial visual state and fails to integrate later transformation evidence, yielding a confident but incorrect response. We refer to W1 by reviewer 1fAq for more details.
>
> Minor comments
> > Thanks for the constructive advice! We will carefully revise our manuscript and incorporate these suggestions in the final version.

---

> > ### Author Rebuttal · Reviewer_LfSu · 2026-04-03
> >
> > Dear Authors,
> >
> > I appreciate your clarifications. I retain a few questions however, which I outline below:
> >
> > To Q2, it makes sense to me that models may be biased by for example the text inputs here. I think the Figure is still very confusing, you write that you show "distribution change" but the text says "In the Empty Image Control (Panel A),  hen the actual conservation task required a ”Conserve” answer, 85.7% of responses remained ”Conserve” under empty images. Critically, when the actual task required ”Not Conserve” answers, models overwhelmingly switched to ”Conserve” responses—71.5%". Maybe I misunderstand your setup here but you should be querying the models with exactly the same empty images and identical prompts here, right? It should not matter whether the "actual task required ”Not Conserve” answers" because the model here should only have access to the exact same information, so why are the numbers different?
> >
> > To Q3, aren't you already showing the median since those are boxplots? Showing the mean additionally seems unnecessary to me and it is visually quite distracting. In general, the Figure is quite small and should be enlarged to improve readability.
> >
> > To L1, I appreciate you running more analyses, but I have no way of understanding them or their significance if you don't offer up more explanation. I understand that is not easy to do in the confines of a 5000 character rebuttal, but as it stands now I can not make sense of them.

---

> > > ### Author Response · Authors · 2026-04-04
> > >
> > > Thank you very much again for your continued engagement with our work.
> > >
> > > To Q2:
> > > > Thank you for raising the question. We will first explain the figure. Both panel A and B are similar. The y axis is the prediction of models with full visual input while the x axis is the prediction of models with empty image or no image input. The i th row and j th coloum indicates the percentage of sampels where model predicts the i th response with full visual input and j th response with empty image or no image input. For instance, the top left 85.7% in panel A means that 85.7% of the samples where model predicts "conserve" in full visual input remains predicting "conserve" when the models are given empty image. The 1st row and 2nd column means that 8.1% of the samples where model predicts "conserve" in full visual input switch to "more" when the models are given empty image.
> > >
> > > > Both panel A and B reflects a large number in (2nd and 3rd row of column 1), meaning that, when the input to model switch from real visual input to empty (blank)  image or no image at all, model switch to predicting "conserve" most of the time (~70% of the time), reflecting models' bias toward selecting "conserve", even if they have no visual input at all. To clarify, models recieve exactly same prompt of all conditions in this experiment. The only alternation is the visual input. We will revise this part further to enhence its clarity for better understanding.
> > >
> > > To Q3:
> > > > Thanks for this suggestion. We will remove it and enlarge the figure in the revision; the additional page in the camera-ready version will accommodate this.
> > >
> > > To L1:
> > > > Confidence Analysis. Our central question is: when the model makes a mistake, is it simply confused and guessing, or is it confidently following the wrong idea? To answer this, we measured how sure the model is when it gives an answer, using the difference in log-probability between the model's most-likely and second-most-likely answer choice at the moment it produces its final answer---a larger gap means the model is more committed to its top choice. We compared four cases: correct conserving predictions (C_correct), correct non-conserving predictions (NC_correct), incorrect non-conserving "same" predictions (NC_same_error), and incorrect non-conserving "different" predictions (NC_other_error). The key finding (https://imgur.com/a/LiRw1Gm): NC_same_error is more confident than NC_correct---when the model incorrectly says "same" on a non-conserving trial, it does so with greater certainty than when it answers correctly. This rules out the benign interpretation that failures are uncertain guesses. Instead, the model appears to have a strong built-in tendency toward "same," which it expresses confidently even when the correct answer is "different."
> > >
> > > > Attention Analysis. We then asked: what is the model looking at when it makes this mistake? Each video has 7 frames, and modern VLMs encode each frame as a grid of visual tokens---where each token is a vector representation of a small spatial patch of that frame. We sum the attention weights over all patches within each frame at the moment the model produces its final answer, then normalize across all 7 frames, yielding a single number per frame per layer that reflects the relative proportion of visual attention devoted to each temporal moment in the video (https://imgur.com/a/uOGZmtM). The key finding: NC_same_errors place substantially more attention on Frame 1 (the initial state) and less on later frames than NC_correct cases, particularly in late layers where decision-relevant processing occurs. In simple terms, the model gets anchored on the starting picture and does not sufficiently use the later evidence where the important change happens. Critically, this Frame-1 anchoring bias is absent for NC_other_errors and conserving-condition errors---it is not a generic property of incorrect predictions, but a pattern specific to this failure mode.
> > >
> > > > Together, these analyses tell a coherent story: on non-conserving trials, the model anchors on the initial visual state and fails to update its representation after the transformation, yielding a confident but incorrect "same" response. This points to a fundamental architectural limitation---current VLMs are predominantly trained on static image-text pairs with limited video data, making them very good at recognizing what is in a picture but much weaker at tracking how a scene changes over time. Without mechanisms for maintaining and updating object-state representations across frames, the model effectively treats transformation reasoning as a static comparison problem. Robust solutions will likely require architectures with predictive, state-based visual abstractions---such as JEPA-style objectives or explicit latent state tracking---rather than static semantic features. We will substantially expand this section in the revision.

---

### Official Review · Reviewer_qGb8 · 2026-03-12

**Soundness:** 1
**Presentation:** 2
**Significance:** 2
**Originality:** 3
**Overall Recommendation:** 4
**Confidence:** 5

**Summary:**

This paper introduces ConservationBench to evaluate the ability of 112 Vision Language Models (VLMs) to understand the conservation of physical quantities, such as number, length, volume, and size. The evaluation results reveal a failure across current VLMs, with performance remaining near chance levels compared to a human baseline of 98.35%. The models exhibit a heavy reliance on textual priors favoring quantity invariance, but their performance degrades when actual visual content is provided. Furthermore, the study demonstrates that increasing temporal resolution, using various prompting strategies such as Chain-of-Thought, or scaling up model size, fails to address these deficits. These findings show that current VLMs lack a robust, grounded understanding of physical transformations.

**Compliance With Llm Reviewing Policy:**

Affirmed.

**Final Justification:**

The authors provided a comprehensive and convincing rebuttal, so most of my concerns are resolved. I raise the final score to 4.

**Key Questions For Authors:**

While this paper provides interesting benchmarks, I have several concerns and questions about
1. experimental details,
2. proper counterparts for physical reasoning,
3. discussion about how to improve the model reasoning capability and the correlation with video understanding.

Please refer to the Weaknesses section for more details.

**Limitations:**

yes

**Strengths And Weaknesses:**

## Strengths
- The paper proposes a refined dataset specifically designed to evaluate the understanding of conservation principles.
- The paper provides a robust statistical analysis by evaluating a wide range of 112 different VLMs.

## Weaknesses
- Although an immense number of models were evaluated, the paper primarily reports performance metrics without a deep dive into how specific architectural features affect the results.
- In the Empty Image Control and Text Control experiments shown in Figure 3, an "Unknown" or "I don't know" option should have been included to account for cases where models cannot make a judgment without visual input.
- The current video-based Q&A format requires foundational skills such as video understanding and object tracking before physical reasoning can occur. To isolate whether the lack of performance is truly a deficit in physical reasoning, the following control groups are necessary:
  1. Providing simple physical laws regarding conservation before asking the question to see if models can apply existing knowledge to a given scenario.
  2. Substituting video scenes with text descriptions to determine if models can perform physical reasoning once the situational context is fully understood.
- While the results claim that VLM physical reasoning is extremely poor, there is no discussion on how this capability might be improved.
- It would be beneficial to discuss the correlation between improvements in physical reasoning and general video understanding capabilities.
- It is unclear whether the order of options (A, B, C) was fixed for all examples or shuffled. Since models often exhibit positional bias, it is uncertain if they are choosing options based on understanding or a biased preference for a specific position.

---

> ### Author Rebuttal · Authors · 2026-03-31
>
> W1: How specific architectural features affect the results.
> > While a full arch ablation (w/ training) is beyond scope, our results across 112 models reveal informative patterns. First, scaling shows no predictive power for conservation accuracy (R²=0.019) and accounts for only 24% of variance in non-conserving accuracy (R²=0.239) in Section 4.5.
> > We further conclude three unsucessful arch attempts: (1) omni models underperform VL counterparts on non-conserving accuracy (Qwen-2.5-Omni-7B: 14.11% vs. Qwen-2.5-VL-7B-Instruct: 21.55%; Qwen-2.5-Omni-3B: 3.06% vs. Qwen-2.5-VL-3B-Instruct: 14.25%), suggesting broader modality (i.e., audio) has no benefit; (2) unified models is near-zero strict accuracy (Janus-Pro-7B: 2.74% strict; Janus-1.3B: 0.66% strict), indicating joint objectives does not help (3) thinking consistently underperform instruct counterparts (Qwen-3-VL-32B-Thinking: Non-Conserve 32.44%, Strict 20.23% vs. Qwen-3-VL-32B-Instruct: Non-Conserve 52.06%, Strict 31.15%), showing explicit reasoning does not compensate for the underlying visual deficit. These collectively suggest arch bottlenecks beyond scale or specific training paradigms.
>
> W2: An "Unknown" or "I don't know" option should be included.
> > We thank the reviewer for this suggestion. We re-ran the Empty Image and Text Control experiments on a subset of n=10 models with an added "I don't know" option. The results are in **https://imgur.com/a/NCYEmYK**.
> The positive correlation between conservation task accuracy and control condition accuracy remains robust (Empty Image: r=0.591, p<0.0001; Text Control: r=0.362, p<0.0001), consistent with our original results (n=62: r=0.678 and r=0.475), suggesting that both visual and textual bias are robust even with third option included.
>
> W3: Two additional control groups.
> > (1): We evaluated models with explicit definition of conservation before the question ("This question is about conservation, which is the understanding that physical quantities remain invariant under transformation despite changes in appearance. Now, answer:")---on a subset of 12 models under the 7-frame, uniform sampling setting. Results are in **https://imgur.com/a/tikXuOY**.
> > Results are consistent with prior experiments: conservation accuracy remains high (Number: 0.863, Length: 0.795, Volume: 0.681, Size: 0.742) while non-conserving accuracy remains near the floor (0.014/0.053/0.128/0.046), consistent with performances without concept-based prompting. This shows that providing explicit conceptual scaffolding does not enable models to flexibly resolve conservation tasks.
> >
> > (2): We evaluated two additional conditions on a subset of 12 models: caption-only and video+caption. Captions are generated by Qwen2.5-VL-72B without answer-leaking instructions and with human verifications. Results are in **https://imgur.com/a/1XRqaJg** and **https://imgur.com/a/KGdciTx**.
> > Both caption-only and video+caption lead to domain-dependent variations: Number and Length retain the invariance bias (Conserve: ~0.78/0.85; Non-Conserve: ~0.06/0.11), while Volume and Size flip to a non-invariance bias (Conserve: 0.014/0.044; Non-Conserve: 0.798/0.684), suggesting captions describing continuous transformations trigger a perceptual-change heuristic. Crucially, neither condition approaches ideal performance that is balanced across domains, demonstrating that the failure persists even when full situational context is provided in text---the deficit is in physical reasoning itself, not visual processing alone.
>
> W4: No discussion on how this capability might be improved.
> > Please refer to our response to L1 from reviewer qGb8. Briefly, our analyses suggest failure is unlikely to be remedied by prompting or augmentation alone, instead pointing toward architectural solutions that enable stable, transformation-invariant representations, such as predictive world models (e.g., JEPA-style) or explicit state-transition encoders.
>
> W5: correlation with physical reasoning and video understanding
> > ConservationBench is designed to isolate transformation-invariant reasoning under controlled conditions, whereas existing benchmarks, such as PhysBench, embed physical understanding within richer, contextual scenarios that introduce confounding factors; we view these as complementary. To further address the concern, we perform a correlation analysis between ConservationBench and other benchmarks using models' performance on all these benchmarks (**https://imgur.com/a/Orz9B5v**). The result shows a strong correlation, indicating that ConservationBench captures a general factor of physical reasoning and video understanding transferable to naturalistic settings, while its controlled design ensures attributing failures specifically to transformation reasoning.
>
> W6: order of options.
> > Thanks for pointing this out. The order in our exps is already shuffled and counterbalanced to eliminate positional bias as a confound. We will document this more explicitly in the revision.

---

> > ### Author Rebuttal · Reviewer_qGb8 · 2026-04-04
> >
> > W4 directs me to 'the response to L1 from reviewer qGb8,' but I am reviewer qGb8. I would like to see the intended response before finalizing my assessment.

---

> > > ### Author Response · Authors · 2026-04-04
> > >
> > > We apologize for the typo. For W4, We meant to direct you to response to L1 from Reviewer LfSu.
> > > Given this response, we will directly reply here:
> > >
> > > To W4:
> > > > We agree that documenting a failure alone is not sufficient, and our goal is not merely to show that current VLMs underperform on yet another benchmark. Rather, we use this task to expose a more fundamental limitation in the current VLM paradigm: these models do not appear to build stable visual representations of objects and their state changes under transformation.
> > >
> > > > Our contribution is therefore not only empirical, but diagnostic: through controlled exps, we isolate the failure mode more precisely than prior work. The issue is not simply low accuracy but show systematic heuristic reversals, and empty-image / text-only controls further show that textual priors often dominate while visual evidence fails to correct them. This suggests that the core limitation is not lack of knowledge, but cannot reliably extract and maintain the relevant object-state representation from visual input.
> > >
> > > > We exmain **Internal Mechanisms** to reveal how model makes mistakes below:
> > > >
> > > > Confidence Analysis. Our central question is: when the model makes a mistake, is it simply guessing, or is it confidently following the wrong idea? To answer this, we use the difference in log-probability between the model's most-likely and second-most-likely answer choice. A larger gap means the model is more committed to its top choice. We compared four cases: correct conserving predictions (C_correct), correct non-conserving predictions (NC_correct), incorrect non-conserving  predictions (NC_same_error), and incorrect non-conserving predictions (NC_other_error). The key finding (**https://imgur.com/a/LiRw1Gm**) is: NC_same_error is more confident than NC_correct, i.e., when the model incorrectly says "conserve" on a non-conserving trial, it does so with greater certainty than when it answers correctly. This rules out the benign interpretation that failures are uncertain guesses. Instead, the model appears to have a strong built-in tendency toward "conserv", which it expresses confidently even when the correct answer is "different."
> > >
> > > > Attention Analysis. We then asked: what is the model looking at when it makes this mistake? Each video has 7 frames, and modern VLMs encode each frame as a grid of visual tokens---where each token is a vector representation of a small spatial patch of that frame. We sum the attention weights over all patches within each frame at the moment the model produces its final answer, then normalize across all 7 frames, yielding a single number per frame per layer that reflects the relative proportion of visual attention devoted to each frame (**https://imgur.com/a/uOGZmtM**). The key finding: NC_same_errors place substantially more attention on Frame 1 (the initial state) and less on later frames than NC_correct cases, particularly in late layers where decision-relevant processing occurs. In simple terms, the model gets anchored on the starting picture and does not sufficiently use the later evidence where the important change happens. Critically, this Frame-1 anchoring bias is absent for NC_other_errors and conserving-condition errors---it is not a generic property of incorrect predictions, but a pattern specific to this failure mode.
> > >
> > > > Together, these analyses tell a coherent story: on non-conserving trials, the model anchors on the initial visual state and fails to update its representation after the transformation, yielding a confident but incorrect "same" response. This points to a fundamental architectural limitation---current VLMs are predominantly built on static image encoders and weak temporal aggregation, which may be sufficient for recognition or description, but are not enough for tracking quantity-preserving transformations over time. Without mechanisms for maintaining and updating object-state representations across frames, the model effectively treats transformation reasoning as a static comparison problem. Even if fine-tuning, reasoning-oriented post-training, or narrow in-distribution supervision improves benchmark performance, such gains would not resolve the underlying issue. Conservation-like judgments are visually intuitive and should not require task-specific hacks or specialized exposure to near-identical training examples. Our results instead suggest a different representational foundation: models that learn predictive, state-based visual abstractions rather than only static semantic features. Approaches based on predictive visual modeling, such as JEPA-style objectives, explicit latent state tracking, or world-model-like architectures, may provide a more principled path forward.
> > >
> > > > **We also summarzie a set of unsucessful architecture attempts** in response to W1. Please also see **our response to W3 of Reviewer 1fAq for qualitative analysis and discussion of failure pattern**.
> > >
> > > > We will substantially expand this section in the revision.

---

### Official Review · Reviewer_1fAq · 2026-03-12

**Soundness:** 2
**Presentation:** 2
**Significance:** 2
**Originality:** 2
**Overall Recommendation:** 4
**Confidence:** 3

**Summary:**

This paper investigates whether modern Vision–Language Models (VLMs) can reason about sequential physical transformations in dynamic visual environments. The authors argue that understanding transformations, such as whether certain physical quantities remain conserved despite visual changes, is fundamental for real-world reasoning. To evaluate this capability, they introduce ConservationBench, a benchmark designed to test whether models can recognize conservation properties across sequences of images depicting physical transformations. The benchmark focuses on four quantitative properties: number, length, volume, and size. Each task requires models to integrate visual information across time and determine whether a property remains conserved after a transformation. The dataset includes both conserving and non-conserving counterfactual scenarios, allowing systematic evaluation of whether models truly understand conservation principles or rely on superficial correlations.

**Compliance With Llm Reviewing Policy:**

Affirmed.

**Final Justification:**

I originally recognized that the paper tackles a meaningful problem with a well-designed benchmark and robustness checks across evaluation settings. My main concerns were largely: 1. No analysis of why models fail (attention patterns, representations), 2. Synthetic-only scenarios limit generalizability, 3. Binary classification tasks are too coarse, 5. Insufficient positioning relative to prior physical reasoning benchmarks, and 6. Only pretrained models evaluated; SFT and world models are missing. During the rebuttal, the author has provided additional justification with new results that address a major part of my concerns. As a reflection, I have decided to increase my score by 1 point.

**Key Questions For Authors:**

Please see weaknesses

**Limitations:**

yes

**Strengths And Weaknesses:**

### **Strengths**
- The paper addresses a fundamental capability required for embodied intelligence and physical reasoning. Evaluating whether VLMs understand invariance under transformation is practically important.
- The proposed ConservationBench is carefully constructed to test conservation across multiple properties (number, length, volume, size) and includes paired conserving vs. non-conserving scenarios. This design helps isolate whether models rely on shortcuts.
- The authors explore multiple factors such as prompt design, frame sampling strategies, and task formulations. This helps demonstrate that the observed failure is robust across evaluation settings.
- The paper provides a strong empirical result: current VLMs do not demonstrate systematic conservation reasoning, suggesting a gap between visual understanding and physical reasoning.
### **Weaknesses**
- While the paper shows that models fail on conservation tasks, the analysis of why this happens is relatively limited. For example, it would be useful to examine attention patterns and intermediate representations to better understand the failure modes.
- Although the benchmark is carefully constructed, all scenarios appear synthetic or controlled. It is unclear how well the findings transfer to more naturalistic video data or real-world tasks.
- The tasks largely reduce reasoning to classification (conserved vs. not conserved). More nuanced evaluation (e.g., reasoning traces and intermediate outputs) might reveal additional insights.
- The relationship between ConservationBench and prior benchmarks for physical reasoning could be discussed more thoroughly.
- The paper mainly evaluates pre-trained models. It would be interesting to see some results on models after SFT and world models that are trained to understand causal visual dynamics.

---

> ### Author Rebuttal · Authors · 2026-03-31
>
> W1: Examine the attention patterns and intermediate representations to better understand the failure modes
> > Great advice! We examine the confidence and attention pattern of Qwen2.5-VL-7B-Instruct.
> >
> > **Confidence analysis.** As shown in **https://imgur.com/a/LiRw1Gm**, the dominant failure on NC examples is predicting "same". We measure confidence via the top1-top2 margin at the decision token; a larger margin indicates greater confidence. Strikingly, NC_same_error is substantially *more* confident than NC_correct, indicating that the dominant NC failure is not a low-confidence fallback but a committed incorrect prediction. Meanwhile, NC_correct is less confident than C_correct, suggesting that correct predictions in the atypical non-conserving setting are made with substantially greater uncertainty.
> >
> > **Attention analysis.** To probe where the model looks, we aggregate attention over visual tokens belonging to each of the 7 input frames to obtain a layer-wise frame-attention distribution in **https://imgur.com/a/uOGZmtM**. A clear pattern emerges: on NC examples, errors predicting "same" place substantially more attention on Frame 1 and relatively less on later frames compared to NC-correct cases, particularly in late layers. Crucially, this strong frame-1 bias is absent for NC non-same errors and for conserving-condition errors, ruling out a generic property of incorrect predictions or "same" responses in general.
> >
> > Taken together, these results suggest: on NC examples, the model appears to over-rely on the initial visual state and underweight later transformation evidence, leading to a confident but incorrect "same" response.
> >
> > We emphasize this is a preliminary analysis on a single model; We will test generalizability across model families and verify causal roles through targeted intervention experiments in the final revision.
>
> W2: Although the benchmark is carefully constructed, all scenarios appear synthetic or controlled. It is unclear how well the findings transfer to more naturalistic video data or real-world tasks.
> > Thanks for the concern. While ConservationBench is a controlled benchmark, it contains no synthetic data, i.e., all videos are recordings of real environments with physical manipulations. Second, the controlled design is not a limitation but a deliberate methodological strength to alleviate confounding factors in scientific discoveries. We refer to our response to W4 with a correlation analysis to further address this concern.
>
> W3: More nuanced evaluation
> > We have conducted a qualitative analysis on reasoning traces of models' responses, which will be added to the revision. For e.g, **https://imgur.com/a/CTDSRg4** shows that Claude-Sonnet-4.5, on a non-conserving volume task, carefully compares liquid levels across frames and concludes that the quantities are equivalent. Although this response captures perceptual similarity, it fails to represent the transformation as a continuous physical process in which partial liquid transfer should dominate any static comparison of apparent levels. Rather than reasoning about whether the transfer is complete or partial, the model reduces the task to frame-by-frame perceptual matching. This is precisely the failure mode we reveal: models default to static visual matching instead of constructing transformation-invariant representations of the physical dynamics.
>
> W4: The relationship between ConservationBench and prior benchmarks for physical reasoning could be discussed more thoroughly.
> > ConservationBench is designed to isolate transformation-invariant reasoning under controlled conditions, whereas existing benchmarks, such as PhysBench, embed physical understanding within richer, contextual scenarios that introduce confounding factors; we view these as complementary.
> > To further address the concern, we perform a correlation analysis between ConservationBench and other benchmarks using models' performance on all these benchmarks (**https://imgur.com/a/Orz9B5v**). The result shows a strong correlation, indicating that ConservationBench captures a general factor of physical reasoning and video understanding transferable to naturalistic settings, while its controlled design ensures attributing failures specifically to transformation reasoning. We will incorporate this analysis into the revised manuscript.
>
> W5: Results on models after SFT and world models that are trained to understand causal visual dynamics.
>
> > First, we would like to clarify that all models tested have gone through the SFT stage (not pretrained models). To further address the concern, we evaluated Cosmos Reason, post-trained with physical common sense and embodied reasoning data. Results (**https://imgur.com/a/yzG7CEt**) show a similar trend as observed in other models, showing near-zero non-conserving accuracy on Number (0.003 ) and Length (0.045) and below chance on Volume (0.229) and Size (0.108) on both conserve and non-conserve condiitons.

---

> > ### Author Rebuttal · Reviewer_1fAq · 2026-04-03
> >
> > Thank you for your detailed responses. They address all my concerns, and I look forward to seeing these new results in the camera-ready version. I will update my score accordingly.

---

> > > ### Author Response · Authors · 2026-04-03
> > >
> > > Thank you for your supportive feedback! We are glad that our rebuttal addressed your concerns, and we will include the new results in the camera-ready version.

---

### Decision · Program_Chairs · 2026-04-30

**Decision:**

Accept (regular)

**Comment:**

This paper proposes ConservationBench, a dataset designed to evaluate VLMs’ ability to recognize whether physical quantities remain invariant under various transformations. The results show that most VLMs fail to answer these questions consistently.

Reviewers appreciate the contribution of introducing this benchmark, which evaluates an important capability of VLMs. They also value the comprehensive analysis of prompt design, frame sampling strategies, and task formulations. The empirical results are thorough, covering a wide range of VLMs and showing the limitations of existing VLMs.

Some additional points raised by the reviewers could further strengthen the paper:
- Provide deeper analysis of why current models perform poorly on this benchmark and discuss potential directions for improvement.
- The tasks are currently simplified as classification problems; expanding to a broader range of task types could be beneficial.

The authors have provided some preliminary results addressing the points above.

AC agrees with the reviewers’ assessment that the proposed benchmark, with its high-quality examples targeting a unique capability of VLMs, makes a meaningful contribution, while the paper could be further improved with more in-depth analysis.